# What Characterizes Effective Reasoning?
# Revisiting Length, Review, and Structure of CoT

**Yunzhen Feng** [1 2 *] **Julia Kempe** [1 2 †] **Cheng Zhang** [1 †] **Parag Jain** [1 †] **Anthony Hartshorn** [1 †]

## Abstract

Large reasoning models (LRMs) spend substantial test-time compute on long chain-of-thought (CoT) traces, but what *characterizes* an effective CoT remains unclear. While prior work reports gains from lengthening CoTs and increasing review via appended *wait* tokens, recent studies suggest that shorter thinking can outperform longer traces. We therefore conduct a systematic evaluation across ten LRMs on math and scientific reasoning. Contrary to the "longer-is-better" narrative, we find that both naively using longer CoTs and more review behaviors are associated with *lower* accuracy.

As CoT unfolds step by step, token-level metrics can conflate verbosity with process quality. We introduce a graph view of CoT to extract structure and identify a single statistic—the *Failed-Step Fraction (FSF)*, the fraction of steps in abandoned branches—that consistently outpredicts length and review ratio for correctness across models. To probe causality, we design two interventions. First, we rank candidate CoTs by each metric at test time, where FSF yields the largest pass@1 gains; second, we edit CoTs to remove failed branches, which significantly improves accuracy, indicating that failed branches bias subsequent reasoning. Taken together, these results characterize effective CoTs as those that *fail less* and support *structure-aware* test-time scaling over indiscriminately generating long CoTs.

## 1. Introduction

Large reasoning models (LRMs) (Jaech et al., 2024; Rastogi et al., 2025) increasingly exploit test-time compute by generating long chain-of-thought (CoT) traces. Challenging prompts are decoded over hundreds of thousands of tokens. A notable line of work, beginning with S1 (Muennighoff et al., 2025) and reinforced in subsequent papers (Ringel et al., 2025; Jurayj et al., 2025), shows that appending *wait* to the generation to increase test-time compute can improve reasoning performance. However, it is unclear whether such long reasoning traces are desired. Long reasoning traces not only significantly increase the resources for those hosting LRMs but also degrade user experience due to latency, especially for questions that intuitively do not require long reasoning. Moreover, recent studies (Wu et al., 2025b; Hassid et al., 2025; Ghosal et al., 2025; Marjanović et al., 2025) report that shorter thoughts are better, and continuing to append 'wait' can induce oscillatory performance. Furthermore, it remains unclear whether different LRMs exhibit similar reasoning behaviors.

These conflicting findings motivate a systematic re-examination of how lexical and structural properties of reasoning traces relate to reasoning performance. In this work, we evaluate the effectiveness of reasoning traces along multiple dimensions and uncover consistent patterns across LRMs. We analyze ten reasoning models with accessible reasoning traces on tasks spanning math and scientific reasoning (HARP, (Yue et al., 2024), and GPQA-Diamond (Rein et al., 2024)), with the aim of providing systematic insight into what characterizes effective reasoning.

We begin by examining two properties that recent work suggests may drive reasoning performance: CoT length and review behaviors. In the S1 approach, inserting *wait* increases generation `Length` and encourages `Review` behaviors, including checking, verifying, or backtracking prior steps. These `Review` behaviors are shown to be important to reasoning (Gandhi et al., 2025; Chen et al., 2024). Therefore, we first investigate how `Length` and `Review` behaviors lead to reasoning improvement observed in Muennighoff et al. (2025). We define `Review Ratio` as the fraction of `Review` tokens within a CoT to isolate the effect of `Review` from `Length`. Using a conditional correlation analysis to

---

[*]Work done during an internship at Meta. [†]Joint advising. [1]Meta Superintelligence Labs [2]New York University. Correspondence to: Parag Jain <paragjain@meta.com>, Anthony Hartshorn <me@anthonyhartshorn.com>.

*Proceedings of the 43rd International Conference on Machine Learning*, Seoul, South Korea. PMLR 306, 2026. Copyright 2026 by the author(s).

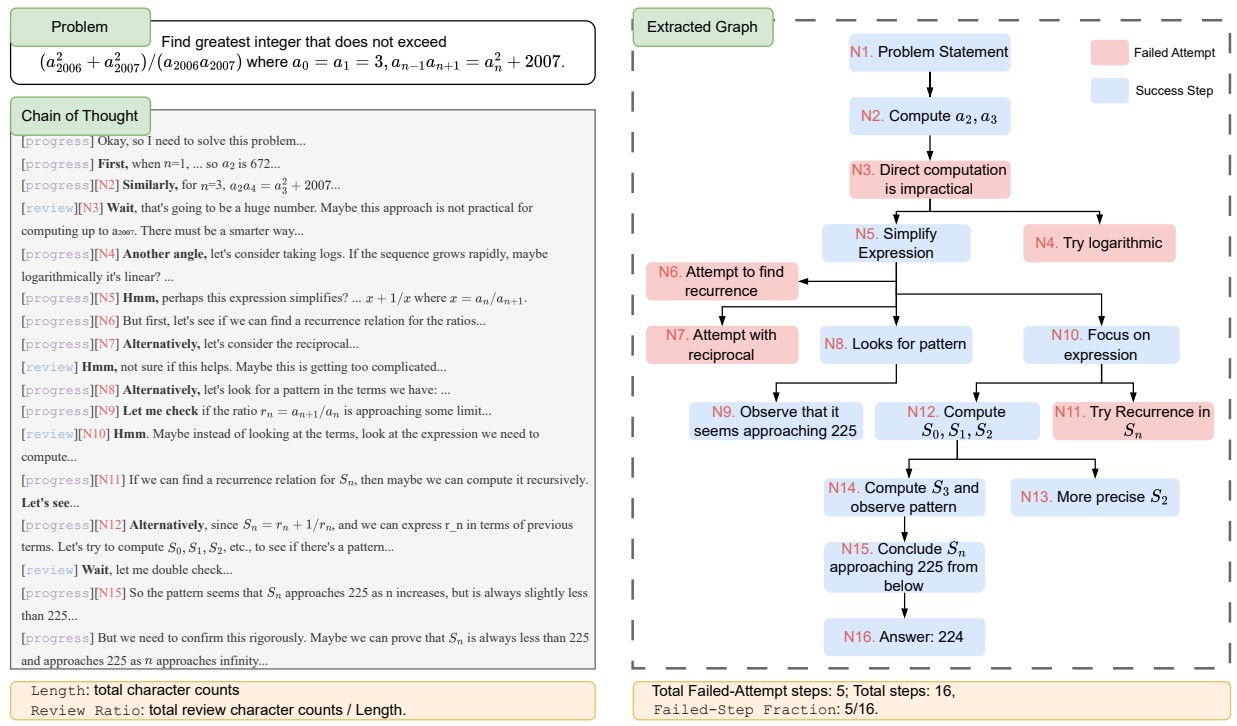

*Figure 1.* **Example.** A chain-of-thought (left) with `Review` annotations and the extracted reasoning graph (right). The CoT is segmented into semantic chunks (Section 3), each labeled `Progress` or `Review`; from these labels we compute `Length` and `Review Ratio`. The right panel shows the graph with nodes ($N_1 : N_{16}$); red nodes denote failed attempts (Section 4). Nodes summarize spans in the CoT, yielding an operational graph representation from which we measure `Failed-Step Fraction`.

isolate the question-level confounding factors, we find consistent patterns across models and datasets. *Within the same question*, shorter reasoning traces are associated with higher accuracy, and lower `Review Ratio` are associated with higher accuracy.

We further hypothesize that `Length` and `Review Ratio` are surface proxies for underlying structural properties of the reasoning (Jiang et al., 2025) and we test one possible cause: the prevalence of failed reasoning branches. We therefore extract a *reasoning graph* for each CoT. This representation allows for the evaluation of graph-level metrics. In particular, we focus on the `Failed-Step Fraction` (FSF): the fraction of steps belonging to failed exploratory branches.

Among graph-level features, FSF emerges as a stronger and more stable predictor of correctness than CoT `Length` or `Review Ratio`, with consistent, significant correlations across difficulty strata and across all ten models on both math and scientific reasoning. These findings support measuring reasoning quality via the reasoning graph. Figure 1 illustrates our annotation and the corresponding extracted reasoning graph.

Finally, we design two experiments to test causality. We first run a test-time intervention on AIME-25 and GPQA-

Diamond: for each problem we sample 64 generations, rerank by each metric, and evaluate top-1 (pass@1) performance. FSF-based selection yields the largest and most consistent gains, with up to 10% accuracy improvement on AIME, while selection by `Length` or `Review Ratio` gives smaller benefits. Second, we intervene on the CoT directly via controlled editing: removing the failed branch substantially increases accuracy on incorrect traces. Together, these results provide causal evidence that FSF is a strong lever for accuracy, long failed branches bias subsequent exploration, and current models do not fully "unsee" earlier mistakes when backtracking.

Our contributions can be summarized as:

- We conduct a wide-ranging conditional correlation test and show that, *within the same question*, longer CoTs and higher `Review Ratio` are negatively associated with accuracy. We measure a stronger correlation for harder questions.

- We introduce a new reasoning-graph extraction method and define the `Failed-Step Fraction`; FSF robustly predicts correctness across models and difficulty strata, outperforming length and review ratio.

- We design a test-time selection intervention providing causal evidence: FSF-based reranking consistently outperforms baseline, Length-, and Review Ratio –based selection on AIME-25 and GPQA-Diamond.

- We directly intervene in CoTs as a causal probe, revealing that failed attempts bias subsequent reasoning; *removing* failed branches substantially improves accuracy.

**Conflict of Interest Disclosure** YF, JK, CZ, PJ, and AH are affiliated with Meta. This paper uses Llama 4 Maverick, a Meta model, as part of the preprocessing pipeline for labeling review behavior. The paper does not evaluate Llama 4 as a target reasoning model in the main benchmark comparisons.

## 2. Related Work

**Scaling Test-Time Compute** Large reasoning models (LRMs) increasingly rely on long, step-by-step CoT traces, reflecting a shift from scaling compute at training time to scaling at test time. This trend is exemplified by OpenAI's O1 series (Jaech et al., 2024) and DeepSeek R1 (Guo et al., 2025), which often generate CoTs with tens of thousands of tokens before providing a final answer. As the number of tokens produced during inference grows, performance tends to improve, exhibiting test-time scaling behavior.

Various research approaches have explored methods to achieve this scaling. Among these, the S1 study (Muennighoff et al., 2025) appends *wait* tokens to increase generation length, prompting the model to continue generating and review its prior reasoning. Follow-up works (Ringel et al., 2025; Jurayj et al., 2025) replace the fixed *wait* token with learned "continue thinking" prompts, reporting larger gains. However, recent work has produced conflicting findings: Wang et al. (2025) show that suppressing *wait* can preserve accuracy while reducing length; Ghosal et al. (2025); Marjanović et al. (2025); Wu (2025) observe that continually adding *wait* initially helps but ultimately degrades performance; and Wu et al. (2025b) provide evidence that longer CoTs are not always better. Furthermore, several results are restricted to small model sets, leaving unclear whether different LRMs exhibit similar behavior. Motivated by these mixed results, we conduct a systematic study across 10 models to understand how Length and Review behaviors generally affect reasoning.

**Extracting Reasoning Structure** To understand the structural properties of reasoning traces, recent work (Jiang et al., 2025; Minegishi et al., 2025) has explored representing CoT reasoning as graphs, where nodes capture reasoning steps and edges represent logical dependencies or flow between steps. Extracting a faithful reasoning graph from an existing CoT is challenging, and only a few recent or concurrent papers attempt it. Jiang et al. (2025) propose a six-round prompt scaffold for summarization, segmentation, and node assignment to extract the graph. Minegishi et al. (2025) instead leverage internal representations: they aggregate sentence-level hidden embeddings, cluster them with $k$-means to form nodes, and connect nodes in the order visited. By contrast, we directly elicit graphs from the model, relying on Graphviz extraction capabilities learned in pretraining and avoiding both multi-call scaffolding and sentence-level embeddings.

**Characterizing Effective Reasoning** Understanding what makes reasoning effective is fundamental to improving LRMs. Guo et al. (2025) showcases an "aha moment," in which the model reviews its previous steps. Subsequent work identifies cognitive behaviors, such as verification and backtracking, as important for reasoning (Gandhi et al., 2025; Hu et al., 2025). However, these behaviors are difficult to measure reliably and previous studies often construct them on synthetic tasks. At the graph level, Jiang et al. (2025); Minegishi et al. (2025) analyze reasoning-tree structure and find these features to matter; Wu et al. (2025a) examines knowledge correctness and information gain. Our primary analysis centers on Length, Review Ratio, and graph-based FSF; additional complementary metrics are provided in Appendices D and F.

## 3. Framework

We pose three research questions: (i) Does increasing CoT length improve reasoning accuracy? (ii) Does increasing Review improve reasoning accuracy? and (iii) What structural properties underlie the effects of length and Review? The first two questions are motivated by ongoing debates surrounding the S1 approach (Muennighoff et al., 2025), while the third seeks to identify more fundamental structural drivers of reasoning performance. In this section, we will outline the framework and define these metrics.

### 3.1. Setup

**Dataset** We leverage the HARP dataset (Yue et al., 2024), which is centered on mathematical reasoning, and the GPQA-Diamond dataset (Rein et al., 2024), which covers scientific reasoning. Both datasets have human-labeled difficulty levels, allowing us to examine patterns across different difficulty strata. The HARP dataset comprises 5,409 math questions sourced from U.S. national math competitions. To reduce computational load, we subsample 50 questions from each of the 6 difficulty levels. We take all 198 questions from GPQA-Diamond.

**Models** We analyze different model families and different model sizes, including both dense models and mixture-of-experts models.

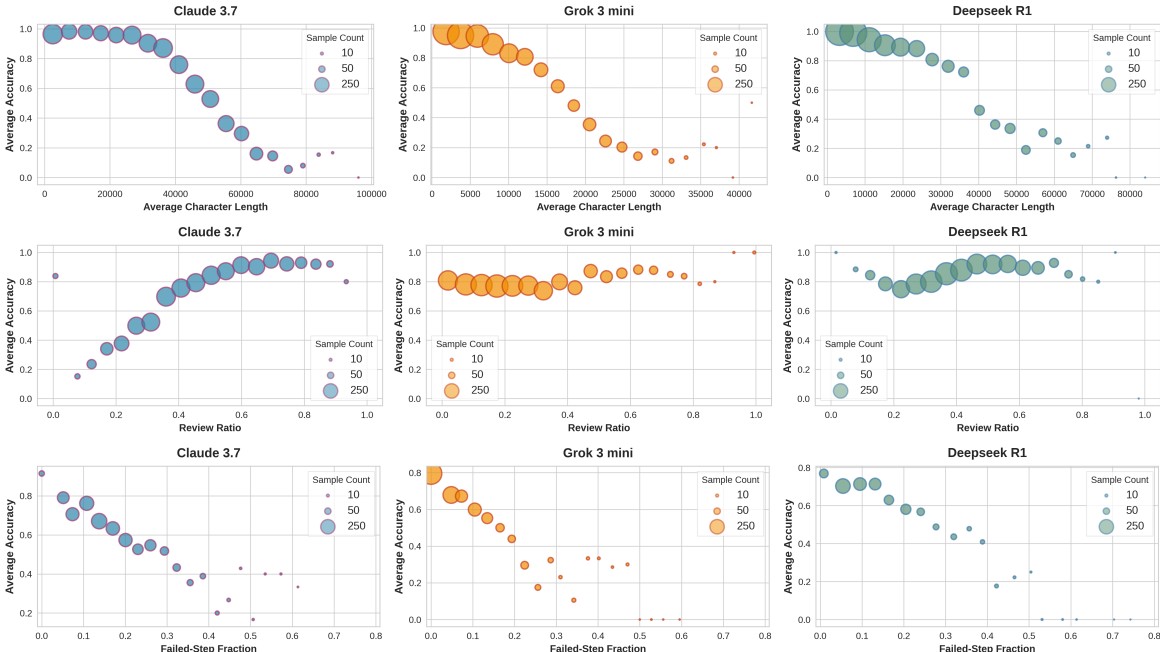

*Figure 2.* Distribution of the metrics—`Length`, `Review Ratio`, and `FSF` and their correlation with accuracy. All measured on CoTs generated for the Level 6 (hardest) subset of the HARP dataset. All three metrics exhibit correlation, with `FSF` the strongest.

*Proprietary models with CoT access*: Claude 3.7 Sonnet Thinking, Grok 3 mini.

*Open-Sourced Families*: Deepseek R1 (20250120), Deepseek Distill Qwen 32B (Deepseek 32B), Deepseek Distill Qwen 7B (Deepseek 7B), Qwen 3 235B, Qwen 3 32B, Qwen 3 8B, GPT oss 120B, GPT oss 20B.

For each question, we generate 16 reasoning traces to ensure that we have enough observations. This allows our analysis to condition on the question to rule out any question-related confounding factors. In total, we analyze 4,800 math reasoning traces and around 3,200 general science reasoning traces for each model.

### 3.2. Metrics

To fairly compare between different models when the tokenizer is different, the following metrics are defined at the character level.

`Length`. We define the CoT `Length` in characters.

`Review`. We measure `Review` behaviors with an LLM-as-a-judge procedure. Each reasoning trace is segmented into chunks using keyword-based heuristics (full list in Table 2). We then prompt the Llama 4 Maverick model (Meta AI, 2025) to label each chunk as `progress` or `review`; the model receives the current chunk together with the preceding five chunks and the subsequent five chunks to provide activity context. We use the following semantics:

*progress*: advances the active reasoning frontier, producing information that later steps rely on.

*review*: reads, checks, restates, deletes, or rewinds existing material without advancing the frontier.

To measure labeling accuracy, we annotate a validation set in-house. We find that Maverick achieves 90% agreement with human labels, with minimal instances of *progress* being mistaken for *review*. Detailed error analysis is presented in Appendix C.1.

With the annotation, we calculate the character-level `Review Ratio` for each reasoning trace: let $s_{t,j}$ denote the $j$-th character in trace $t$ and let $N_t$ be its total number of characters,

$$\texttt{Review Ratio}_t = \frac{1}{N_t} \sum_{j=1}^{N_t} \mathbb{1}[s_{t,j} \text{ lies in a } \texttt{Review} \text{ chunk}].$$

**Reasoning Graph** A CoT naturally unfolds step by step, with steps varying in length and purpose. `Length` and `Review Ratio` are token-level (character-level) measures that can conflate verbosity with process quality. We further extract a reasoning graph for each CoT to probe structural properties. Specifically, we prompt Claude 3.7 sonnet with thinking disabled to convert each CoT into Graphviz format (Gansner, 2012). We view the resulting graph as an operational representation of the trace, rather than a unique ground-truth reconstruction: segmentation, edge

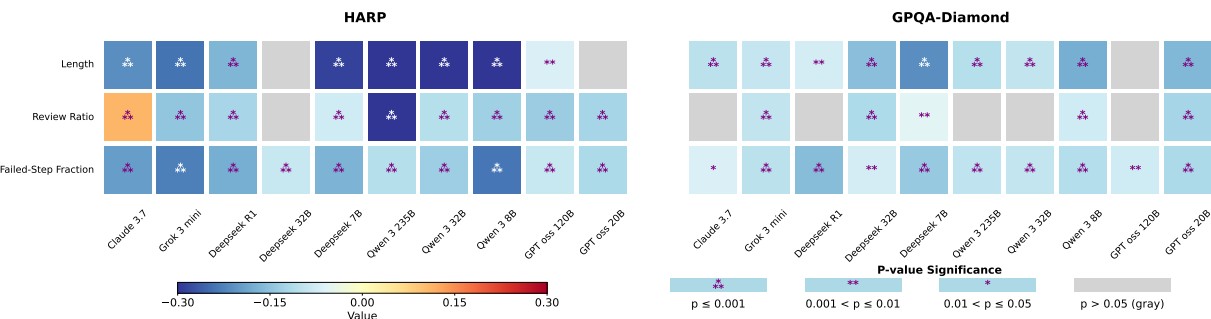

*Figure 3.* Conditional correlations computed on the full dataset. Correlations are shown with a color scale; non-significant cells ($p > 0.05$) are grayed out, and * denotes statistical significance (see the legend). We color * white or purple for visualization only. More colored cells indicate broader prevalence of correlation; darker colors denote stronger correlations. When controlling for question-level confounders, all three metrics correlate with accuracy, with FSF significant across all models and both datasets.

placement, and branch boundaries can be ambiguous in long natural-language reasoning. Figure 1 visualizes one example. Our extraction procedure is simple, avoiding the complex prompting or embedding pipelines of Jiang et al. (2025); Minegishi et al. (2025). The Claude-produced graphs compiled without error in 100% of cases, but compilability is only a syntactic check. We therefore validate the graph-derived measurements functionally: FSF is stable across two extractor models (Appendix D.4), outpredicts alternative graph metrics (Appendix D.2), and supports the intervention results in Section 5.2. Full extraction details and the complete list of graph metrics are provided in Appendix D.

Among graph metrics, we highlight one candidate as a potential structural driver of Length and Review Ratio:

Failed-Step Fraction, the fraction of reasoning nodes in the graph that are marked as failed/abandoned:

$$\text{FSF} = \frac{\text{\# failed nodes}}{\text{\# all nodes}}.$$

During extraction, we ask the model to color-code nodes as successful or failed attempts[1]. This labeling enables a direct computation of the failed-step fraction. In Figure 1, we provide an example CoT with the chunks and annotations of *progress* and *review*, and the extracted reasoning graph. In Appendix D.4, we use a different model to extract the graph and measure the alignment of the results. We observe strong alignment.

Beyond these core metrics, we also evaluate additional graph-based measures (Appendix D) and stylistic features including motivation levels (Appendix C.4), and progressiveness similar to (Wu et al., 2025a) (Appendix F). See the

[1]We emphasize that the "failed attempt" label is local to the reasoning trajectory (e.g., an abandoned branch), not a judgment of step correctness. A CoT may yield an incorrect final answer while every step is labeled "successful"—even when Claude is able to correctly judge the final answer as wrong.

corresponding appendices for detailed definitions.

## 4. Correlation Analysis

We now present our results, beginning with general distribution visualizations, followed by conditional correlation analyses that control for confounding factors.

### 4.1. Metric Distributions

We first visualize the distributions of Length, Review Ratio, and FSF with accuracy in Figure 2 using the HARP Level-6 set (the hardest split). In general, across all three models, shorter CoTs are associated with higher accuracy. For Review Ratio, Claude 3.7 shows a positive trend: higher Review Ratio brings higher accuracy, while the other two models are mixed. For FSF, lower FSF is correlated with higher accuracy, with an approximately linear relationship. However, drawing broad conclusions from raw correlations is risky because of confounding factors. For example, harder questions or specific domains (e.g., algebra) may require longer CoTs and more Review behavior, while also having lower accuracy, which can induce spurious correlations.

To mitigate these confounders, we generate 16 CoTs per question and run a conditional correlation test that conditions on the question level. Specifically, for each metric, we subtract the question-level mean from each of the CoT's value (i.e., include question fixed effects) and then correlate these residualized values with residualized correctness across all data. This controls for question-level heterogeneity and yields reliable estimates. In this correlation analysis, we filter out questions where all generations are correct or all are incorrect, as they provide no signal.

In addition, we fit a Bayesian generalized linear mixed-effects model (GLMM) for correctness as a function of each metric, with random intercepts at the question level. Full model specification and results are provided in Ap-

pendix C.3. The GLMM results align closely with the conditional-correlation analysis: whenever the conditional correlation is significant, the corresponding GLMM coefficient is significant with the same sign. This concordance provides a second line of evidence.

### 4.2. Conditional Correlation Analysis

**Overall Conditional Correlations**   We report conditional correlation results in Figure 3 for all the CoTs on HARP and GPQA-Diamond. Cell color encodes the correlation's sign and magnitude; p value significance is indicated by stars (*** $p \leq 0.001$, ** $0.001 < p \leq 0.01$, * $0.01 < p \leq 0.05$). Cells with p > 0.05 are grayed out, denoting lack of statistical significance. Therefore, more colored cells indicate broader prevalence of correlation; darker colors denote stronger correlations.

For `Length`, we observe a consistent negative correlation across both datasets and most models at the CoT level: shorter CoTs correlate with higher accuracy. For `Review Ratio`, most models similarly exhibit significant negative correlations, with lower `Review Ratio` associated with higher accuracy. Claude 3.7 on math reasoning presents a notable exception, showing the opposite trend as illustrated in Figure 2. These findings provide clarity on the S1 debate and support recent observations in (Wu et al., 2025b; Hassid et al., 2025).

For `Failed-Step Fraction`, we find that FSF correlates significantly with accuracy across every model in both math and scientific reasoning tasks, yielding more consistent correlations than either `Length` or `Review Ratio`. The pattern is robust: lower FSF consistently correlates with higher accuracy – even for Claude, which uniquely benefits from higher `Review Ratio` unlike other models. All these patterns support FSF as the intrinsic driver behind `Length` and `Review Ratio` effects.

**Conditional Correlations by Difficulty Level**   Different questions may require different solution strategies. The human-labeled difficulty level reflects how complex a question is by human standards. We therefore compute conditional correlations within each difficulty level to test whether the correlation holds across difficulty strata. Results are shown in Figure 4. We omit Level 1 in HARP and the Postgraduate level in GPQA-Diamond because the correlation test in these strata includes fewer than 100 CoTs.

We observe distinct patterns when stratifying by question difficulty across both datasets. On HARP, correlations are most consistently significant on harder items (levels 4, 5, and 6) for all three metrics. This concentration is intuitive: for easier questions, models can succeed along multiple trajectories, weakening metric-accuracy correlations. Correspondingly, on easier questions, we see mixed patterns, with

some Deepseek-class models occasionally benefiting from higher `Review Ratio` and longer `Length`. Within GPQA, we find consistent patterns across the Hard Undergraduate and Hard Graduate splits. `Length` remains a prominent predictor, while `Review Ratio` shows less consistent significance within difficulty bands, aligning with the weak `Review Ratio` effects on GPQA shown in Figure 2. Notably, while Claude 3.7 shows no significant correlation across all GPQA data, it does exhibit correlation within the Hard Graduate split, demonstrating that difficulty-specific patterns can be masked in aggregate analyses.

Across both datasets, FSF demonstrates the strongest and most consistent performance. When significant correlations emerge, FSF exhibits consistent negative correlations across all models and difficulty levels, with more significant correlations than either `Length` or `Review Ratio`. This further supports FSF as the key structural metric. In summary, we highlight the following observation:

> Across CoTs for the same question, shorter `Length`, lower `Review Ratio`, and lower FSF all generally correlate with higher accuracy, with more pronounced effects on harder math questions. FSF stands out as the strongest and most consistent predictor.

Beyond FSF, we evaluate correlations for additional graph-based metrics, which show consistently weaker effects than FSF and are mostly significant only on math reasoning (full results in Appendix D). We also examine stylistic features including motivation levels, review positions, and progressiveness entropy (Appendices C.4 and F). These analyses reveal that models exhibit distinct generation styles, but these stylistic features fail to correlate consistently with accuracy across models. These model-dependent behaviors would introduce bias when comparing metrics across models, reinforcing our methodological approach: estimating correlations within each model, then identifying patterns that replicate across models.

## 5. From Correlation to Causality

Having established correlations between `Length`, `Review Ratio`, `Failed-Step Fraction`, and correctness, we now ask whether these correlations hold causally. We design two experiments: first, test-time selection (Section 5.1); second, controlled CoT editing targeting FSF (Section 5.2).

### 5.1. Test-Time Selection

We now use test-time selection as a causal probe. Beyond correlations, we ask whether a metric leads to higher accuracy when it serves as the rule that picks the best final

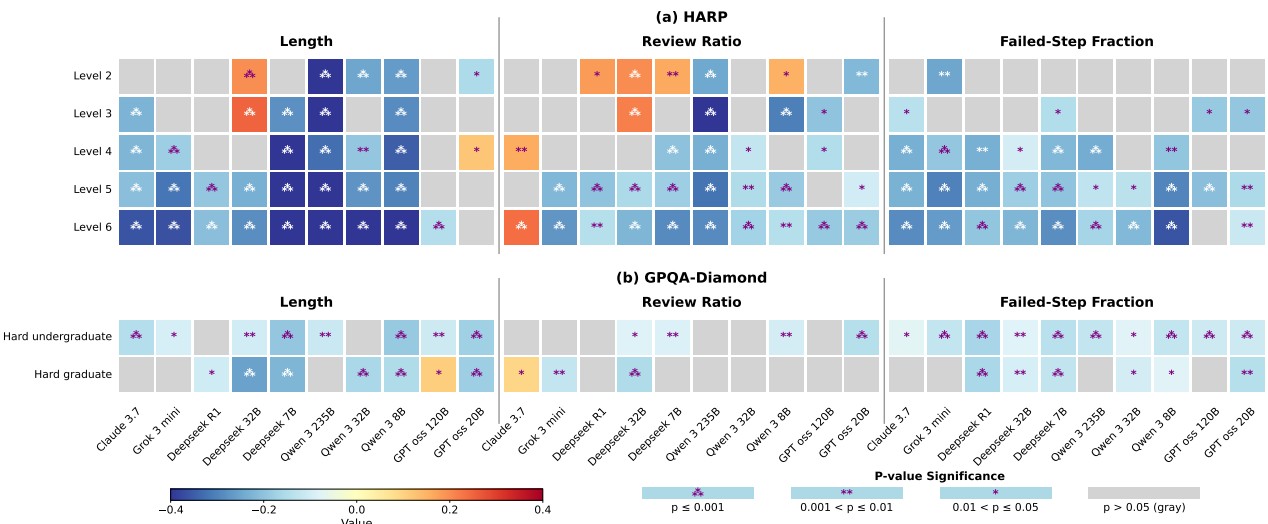

*Figure 4.* Conditional correlation by human-labeled difficulty level for generated CoTs. Top: HARP; bottom: GPQA-Diamond. Same plotting and legend as Figure 3. In math reasoning, correlations are stronger for harder questions.

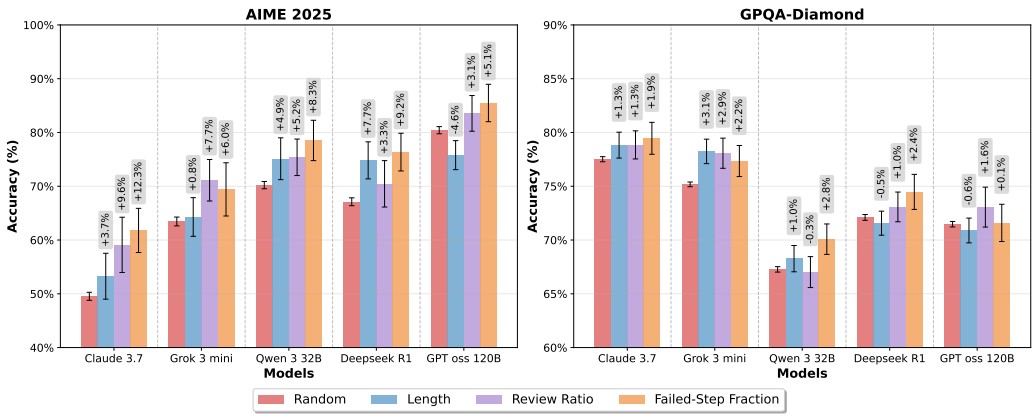

*Figure 5.* Pass@1 with test-time selection by `length`, `Review Ratio`, and FSF. Error bars show bootstrap standard deviations. FSF generally achieves the largest gains, supporting its role as a causal lever.

answer. For each question, we hold the candidate set fixed (same model and decoding) and intervene on the selection policy: we re-rank candidates by the metric and take the top-1. This intervention changes only the distribution of the final selected output. A strong metric should preferentially select correct solutions, yielding the highest pass@1 under this intervention.

**Setup** We evaluate on AIME 2025 (30 problems), which is widely regarded as contamination-free math dataset for recent LRMs, and on the full GPQA-Diamond set. For each problem and model, we sample 64 independent generations. For a given metric, we rank the 64 candidates and compute pass@1 from the top-1. We compare four selectors: (i) FSF (lower is better), (ii) `Length` (shorter is better), (iii) `Review Ratio` (lower is better, opposite for Claude 3.7), and (iv) random selection. Since pass@1 can be noisy in this regime, we estimate uncertainty via bootstrap: for each

model–problem, we draw 200 replicates by resampling the 64 candidates with replacement, re-rank by each selector, record top-1 accuracy, aggregate over problems, and report the mean and standard deviation across replicates. Results are shown in Figure 5.

**Results** We observe that FSF is the strongest selector across models and datasets. On AIME 2025, choosing the single best candidate by FSF yields gains of roughly 5–13% over the random baseline. FSF delivers consistent and significant improvements for all models. `Review Ratio` and `Length` also improve accuracy for most models, with the exception of `Length` on GPT oss 120B. On GPQA-Diamond, FSF again produces significant gains for every model. These interventions provide *causal evidence* for all metrics, with the strongest and the most consistent effect for FSF. Notably, FSF is estimated by Claude 3.7, the weakest model on math reasoning in Figure 5, without access to the ground truth

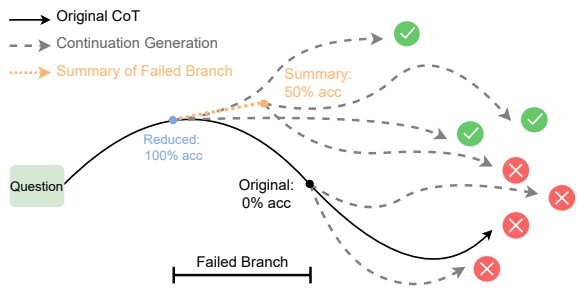

*Figure 6.* Visualization of the continuation generation setup. For incorrect CoTs, we either remove the failed branch or append a brief summary, then evaluate accuracy by continuing from the partial CoT (gray dashed arrows). Table 1 are reported for three setups: *reduced* (failed branch removed), *original* (failed branch retained), and *summary*.

answers, yet it still yields consistent accuracy gains for all models. In this design, we do not rely on a strong judge, we do not provide ground truth answers, and we only ask the model to extract the graph (not to label correctness), which minimizes the risk of knowledge leakage. When Claude 3.7 both generates and estimates FSF and then selects by it (self generate, self estimate, self select), accuracy improves by up to 12% for math.

## 5.2. Modifying the CoT

In this section we further investigate: Why does higher FSF harm performance? Within the correlation test in Section 4.2, we further examine whether the depth of the first failed-step correlates with correctness. The result is included in Figure 8 as 'First Failed Step Depth'. We find little to no correlation across all models. This suggests that it is the presence and extent of failed attempts, rather than when they occur, that harms performance. This observation motivates the following controlled edit: would removing failed exploration improve the accuracy?

To do so, we must first identify where each failed exploration starts. Specifically, when extracting the reasoning graph with Claude 3.7, we also ask it to identify where a failed branch begins (full prompt in Appendix E). We then remove that branch, from its start through the failed attempt steps, and evaluate how its removal changes the accuracy of the partial CoT. We apply this procedure to 80 incorrect HARP traces generated by Deepseek R1 and 160 incorrect traces generated by GPT oss 120B. We compare three variants, each evaluated at both the first and the last failed branch (six settings total): (1) the original reasoning prefix containing the failed branch, with all subsequent steps truncated; (2) the reduced reasoning prefix that includes only the steps up to the failed branch; and (3) the initial prefix plus a concise summary of the failed branch. For each partial CoT in each setting, we perform eight continuation generations to

reliably assess accuracy, without a token-budget limit. We perform 11,520 continuation generations in total. Figure 6 illustrates our CoT-editing procedure and the continuation generation used to probe accuracy.

Table 1 reports the results. For both models, removing the failed branch, at either the first or last failed point, substantially increases the accuracy (the probability that the existing partial CoT reaches the correct answer), by roughly 8–14%. This indicates that the models are capable of producing a successful generation, but the presence of a failed branch markedly lowers that probability. Providing a short summary of the failed branch also improves accuracy, though not as much as removing it entirely. Overall, these results suggest that long failed branches bias subsequent exploration even after backtracking; current models do not fully "unsee" past mistakes. Overall, our findings support *quality-aware test-time scaling*: prefer structure-aware selection (Yao et al., 2023; Bi et al., 2024) and context management with targeted branch pruning/summarization (Snell et al., 2024; Hao et al., 2025; Liao et al., 2025) over indiscriminately generating longer CoTs.

> In conclusion, FSF is the strongest metric that holds causally. Failed branches harm performance by biasing subsequent exploration; removing them improves accuracy.

## 6. Discussion

In this paper, we start with the question: What characterizes effective reasoning? Prior works (Muennighoff et al., 2025; Ringel et al., 2025; Jurayj et al., 2025) suggest scaling test-time compute by inserting *wait* tokens to the generation, lengthening CoTs and encouraging Review behaviors. We therefore test whether Length and Review at test time correlate with correctness at the CoT level. Beyond these token-level proxies, we introduce a method to extract a reasoning graph and study the structural property Failed-Step Fraction. Contrary to the "longer and more Review are better" narrative, we find the opposite: shorter CoTs and lower Review Ratio associate with higher accuracy across math and science. FSF is the strongest predictor, showing significant correlations for all 10 models on both datasets: lower FSF reliably aligns with higher accuracy. Correlations alone don't show mechanism, so we run two causal tests: (1) test-time selection using FSF and (2) targeted editing that removes failed branches from the CoT. Both interventions significantly improve performance and support FSF as not only predictive but also causal: models can reach correct answers, but failed branches bias subsequent exploration and reduce success.

Overall, our work provides important insights into test-time

*Table 1.* Accuracy reported as mean $\pm$ standard deviation (in %). We edit the CoT by deleting failed branches or replacing them with summaries, and measure the effect on accuracy using 8 continuation generations per CoT. Edit performed on a subset of incorrect CoTs from HARP. Removing the failed branch significantly improves the accuracy.

| Model | | Original | Reduced | Reduced with Summary |
|---|---|---|---|---|
| Deepseek R1 | First Failed Branch | 20.89% ($\pm$1.36%) | 29.42% ($\pm$1.66%) | 28.14% ($\pm$1.38%) |
| | Last Failed Branch | 9.72% ($\pm$0.98%) | 23.75% ($\pm$1.36%) | 22.57% ($\pm$1.37%) |
| GPT oss 120B | First Failed Branch | 28.05% ($\pm$0.85%) | 36.41% ($\pm$0.95%) | 29.51% ($\pm$0.90%) |
| | Last Failed Branch | 16.50% ($\pm$0.71%) | 27.33% ($\pm$0.85%) | 25.22% ($\pm$0.89%) |

scaling. We highlight FSF as a robust indicator of effective reasoning. Rather than simply scaling token count, structural quality—specifically controlling failure propagation in context—appears more impactful. As test-time scaling rises, quality-focused strategies (managing failed exploration via context control (Liao et al., 2025; Team et al., 2025)) complement quantity-based approaches.

## Impact Statement

This work studies how structural properties of chain-of-thought traces relate to model accuracy. The intended impact is to improve understanding of test-time reasoning behavior and to encourage more efficient, structure-aware inference rather than indiscriminately increasing generated tokens. Potential risks include overinterpreting extracted reasoning graphs as faithful explanations of a model's internal computation, or using post-hoc trace metrics as a substitute for task-specific validation. We therefore present FSF as an operational measurement of generated traces, discuss extraction limitations, and recommend interpreting it alongside accuracy and robustness checks. More broadly, methods that reduce unnecessary reasoning length may lower inference cost and latency, but should be evaluated carefully in high-stakes applications before deployment.

## Ethics Statement

During this project, we only leverage Large Language Models to help polish and aid some of the writing, and aid some of the coding. All writeups and codes are then carefully scrutinized by the authors.

## Reproducibility Statement

We aim to provide full details for reproducing our results. We generate CoTs on public math and scientific reasoning datasets; all generation settings and prompts appear in Appendix B.1, including details for test-time selection and continued generation in the intervention tests. Metrics are annotated via LLM-as-a-judge: definitions of Length, Review Ratio, and Failed-Step Fraction are in Section 3.2; definitions of additional stylistic features (e.g., Motivation) are in Appendix C.2; definitions of related graph features are in Appendix D.2; and graph-extraction prompts are in Appendix D. Review Ratio and some stylistic features rely on semantic chunking; the keyword list is in Table 2. We probe correlations using conditional correlation and a generalized linear mixed model, detailed in Section 4.1 and Appendix C.3, respectively. In the intervention tests, test-time selection uses the metric to pick the top-1 generation. For the controlled editing experiment, Appendix E.2 provides prompts to align graph nodes with the CoT and to identify where failed exploration begins. We are unable to release the original codebase at this time, but include the prompts, metric definitions, validation checks, and generation settings needed to reproduce the pipeline.

## Acknowledgment

The authors would like to specially thank Kunhao Zheng, He He, Akhil Mathur, Avraham Ruderman, Pu Yang for valuable discussion and helpful insights. We would also like to thank Alan Schelten, Anirudth Goyal, Dulhan Jayalath, Graeme Nail, Hengyuan Hu, Jelmer van der Linde, Lovish Madaan, Nikolay Bashlykov, Qinqing Zheng, Richard Pang, Sam Devlin, Shuangrui Ding, Tatiana Shavrina, Zheng Zhao for discussion. The authors would also like to thank Zebing Lin, Andrii Chernukha, Guillermo Terrazas, and Jacob Logan for infrastructure support. YF and JK acknowledge support by the Simons Foundation through the Collaborative Grant "The Physics of Learning and Neural Computation".

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

# A. Other Related Works

Golovneva et al. (2023) propose a suite of metrics for step-by-step reasoning (e.g., alignment, hallucination, commonsense), computed from sentence-level embeddings. Their analysis targets models producing relatively short CoTs; extending these embedding-based metrics to modern LRMs that generate very long CoTs (tens to hundreds of thousands of tokens) is nontrivial and computationally burdensome. Consequently, it is unclear how to apply that framework in our setting.

Related to length, the efficiency of CoT reasoning (Zhang et al., 2025; An et al., 2025; Gao et al., 2025) is another desirable property since we prefer correct solutions achieved with minimal computation. However, efficiency is only well defined for *correct* CoTs and is ambiguous for incorrect ones; accordingly, we do not analyze efficiency in this paper and focus instead on metrics applicable to both correct and incorrect traces.

# B. Details on the Generations

## B.1. Models and Prompts

We leverage all these models:

*Proprietary models with CoT access*: Claude 3.7 Sonnet Thinking (Anthropic, 2025), Grok 3 mini (xAI, 2025),

*Open Sourced Families*: Deepseek R1 (0120), Deepseek Distill Qwen 32B (Deepseek 32B), Deepseek Distill Qwen 7B (Deepseek 7B), Qwen 3 235B (Yang et al., 2025), Qwen 3 32B, Qwen 3 8B, GPT oss 120B (Agarwal et al., 2025), GPT oss 20B.

For all the reasoning models, we generate 16 responses with 4 temperature, 0.3, 0.6, 0.8, and 1.0. The top p is always set to be 0.9. Claude 3.7 Thinking only allows generation with temperature 1.0, so we use 1.0 for all the generation. For GPT oss 120B and GPT oss 20B, we use the medium thinking mode.

The prompt used for AIME and HARP is:

> Solve the following math problem efficiently and clearly. Please reason step by step, and put your final answer within $\boxed{answer}$.
> Where [answer] is just the final number or expression that solves the problem.
> Problem: {Question}

The prompt used for GPQA-Diamond is:

> What is the correct answer to this question:
> {Question}
> {Choices}
> Format your response as follows: "The correct answer is (insert answer here)".

In the experiment of continuation generation, we follow the suggested optimal temperature 0.6 for all models. For test-time selection, we generate 64 CoTs similarly, using four temperatures with 16 CoTs per temperature.

Across all evaluations, we define the CoT as the text between `<think>` and `</think>` (or the model-specific equivalents) for all ten models. All annotations and evaluations are performed exclusively on this thinking portion.

## B.2. Evaluation

We use the Math-verify package (Kydlíček, 2025) to evaluate the correctness for math reasoning.

For GPQA-Diamond, we parse outputs using the answer template. Because some smaller models do not consistently follow the required format in the prompt, we augment the parser with additional templates to robustly extract the final answer, ensuring correlations are computed against true answer correctness.

## B.3. Keywords for Chunking

We report the full list of keywords used for chunking in Table 2.

| Keywords | | |
|---|---|---|
| Wait | Let me step back | Hang on |
| Hold on | Let me double check | Hold on a minute |
| Hold on a second | Am I missing something | Alternatively |
| Instead | Similarly | I'll approach this from another angle |
| Let's explore alternative approaches | Looking at other approaches | Let me check |
| But let's check | But wait | Let's check |
| I should check | Let me verify | Let's verify |
| Another thought | I should double-check | Let me double-check |
| Let me re-examine | Let me confirm | Looking at the options |
| Looking at the answer choices | Let's look at the options | Let's look at each choice |
| Looking at the other choices | Looking at the answer options | Let me just confirm |
| Another angle | Another check | Let's think again |
| Let's also think about | Let me think about | Another point |
| So back to | Another possibility | Let's proceed step by step |
| Looking at the candidate answers | second thought | Let's break it down |
| Let me reconsider | Let's go back | re-analyze |
| re-check | reconsider | re-examine |
| First, | go though each option | another approach |

*Table 2.* Collection of Keywords

## C. Lexical Metrics

We first assess the quality of the `Review` annotations by comparing them with human labels. We then describe the motivation-annotation pipeline in Appendix C.2. Finally, we report the generalized linear mixed-effects model (GLMM) specification and results in Appendix C.3.

### C.1. Alignment with Human Annotations

Recall that we leverage the Maverick model to label each chunk into `progress` or `review`, with the following definition:

*progress*: advances the active reasoning frontier, producing information that later steps rely on.

*review*: reads, checks, restates, deletes, or rewinds existing material without advancing the frontier.

We study how reliable the model's annotation is when acting as a judge—a consideration often missing from LLM-as-judge work. To evaluate this, we collect 30 long reasoning traces from Deepseek R1 and Qwen 3 235B, spanning math and general science, in both free-form and multiple-choice settings. Each trace is segmented into chunks (around 40 per trace on average), and each chunk is labeled by the authors as `progress` or `review`. We then compare these human labels to the model's own annotations.

In this way, we obtain the confusion matrix shown in Table 3, which illustrates the accuracy of the annotation at the character level. When considering `review` as the positive class, the pipeline demonstrates a low type I error, meaning it rarely misclassifies `progress` as `review`. We allow the model to misclassify some `review` as `progress`, as this serves as a lower bound for `review`.

| True\Predicted | review | progress |
|---|---|---|
| review | 53.8% | 10.2% |
| progress | 1.2% | 34.8% |

*Table 3.* Confusion Matrix for `progress` and `review` annotation.

### C.2. Motivation Annotation

Besides `review`, we hypothesize, drawing on insights from cognitive science (Facione, 1990), that `Motivation` is a key feature: whether the model exhibits a clear goal and strong motivation behind each action, especially during reviews. Accordingly, we measure the motivation level for all review chunks labeled in the previous section.

We use the same chunking protocol. For each review chunk, with its preceding 5 and following 5 chunks as context, we ask the model to annotate the current chunk's motivation as `clear`, `semiclear`, or `unclear`. Definitions:

*Clear motivation*: The chunk states a review action (verify / re-check / backtrack / reread. . . ) and cites a specific trigger / rationale for that action, such as a rule number, mismatch, explicit ambiguity, or other concrete evidence.

*Semi-Clear motivation*: The chunk states a review action and gives only a generic reason ("make sure it's correct", "something seems off", "to be safe") with no concrete trigger.

*Unclear motivation*: The chunk shows a review action but gives no stated rationale at all; the motive must not be inferred.

The motivation score is computed at the character level: for each character within `Review` spans, we assign 1.0 for clear motivation, 0.5 for semi-clear, and 0.0 for unclear, then average over all `Review` characters. We defer the correlation results of `Motivation` to Appendix C.4.

### C.3. Correlation with GLMM

Apart from the conditional correlation test, we also leverage a Generalized Linear Mixed Model (GLMM) to learn the correlations. For a metric $m$, GLMM fits the following equation to estimate the effect of $m$ on accuracy:

**GLMM model:**

$$\text{logit}(P(acc_i)) = \beta_0 + \beta_1 m_i + u(\text{question}_i). \tag{1}$$

The GLMM addresses question-level heterogeneity via a question-specific random intercept $u_{\text{question}(i)}$ with a Gaussian prior. Here, $i$ indexes reasoning traces, and logit denotes the logistic link function. We interpret $\beta_1$ as the association (direction and magnitude) between the metric and correctness. The Gaussian prior enables estimating the posterior mean and standard deviation of $\beta_1$. Thus we also derive Wald-style $p$-values to assess significance.

We summarize the GLMM results in Figure 7. Compared with Figure 3, the pattern of colored cells largely matches: whenever the conditional-correlation analysis flags a significant effect, the GLMM yields a coefficient with the same sign and significance. This concordance provides a second line of evidence supporting our findings.

### C.4. Other Metrics

We also evaluate correlations between accuracy and the following metrics:

- `Review Centroid`: the median position of all `review` chunks within a trace, normalized to $[0, 1]$.

- `Review Chunk Fraction`: the fraction of chunks labeled `review` among all chunks.

- `Review→Progress Switch Count`: the number of transitions where a `review` chunk is followed by a `progress` chunk, normalized by the total number of chunks.

- `Motivation Score`: the fraction of `review` chunks that state a clear motivation for the action (details in Appendix C.2).

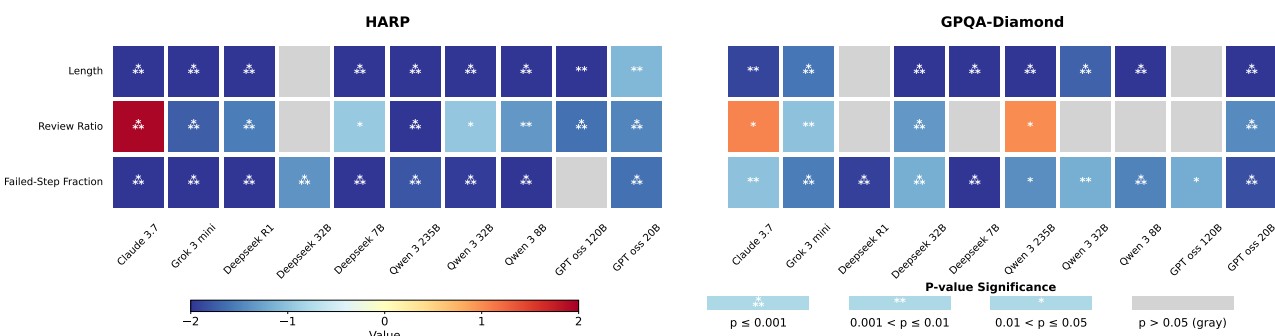

*Figure 7.* GLMM coefficients estimated on the full dataset. Coefficients are shown with a color scale; non-significant cells ($p > 0.05$) are grayed out, and * denotes statistical significance. More colored cells indicate broader prevalence of correlation; darker colors denote stronger correlations. We observe strong alignment with the conditional-correlation patterns shown in Figure 3. This concordance provides a second line of evidence.

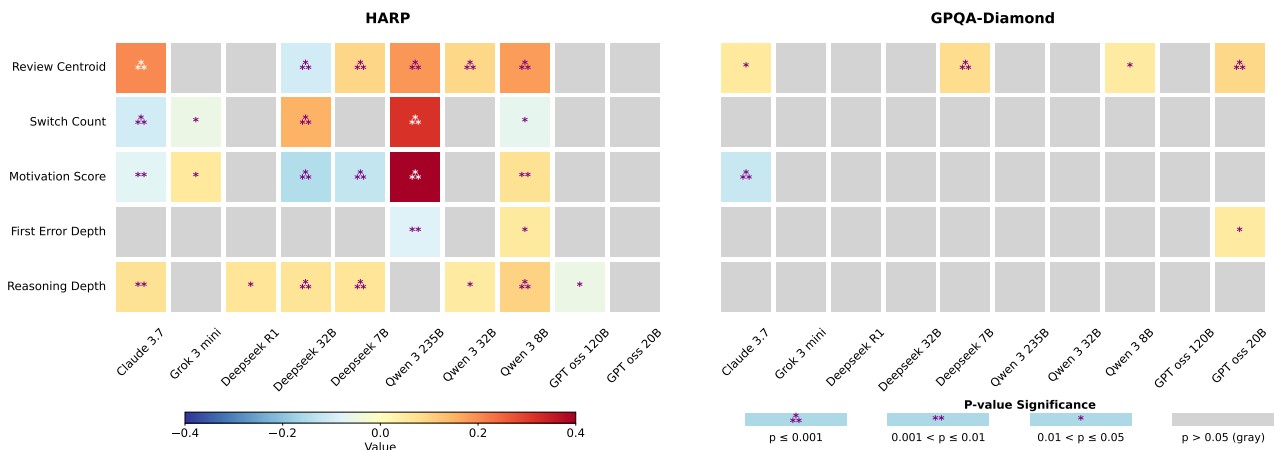

*Figure 8.* Conditional correlations computed on the full dataset, for `Review` position, `Motivation Score`, `First Failed-Step Depth`, and overall `Reasoning Depth`. Again, correlations are shown with a color scale; non-significant cells ($p > 0.05$) are grayed out, and * denotes statistical significance (see the legend). We color * white or purple for visualization only.

- `First Failed-Step Depth`: the depth of the first failed step in the reasoning graph.

- `Reasoning Depth`: the depth of the reasoning graph from the problem statement.

**Results.** Figure 8 reports the correlations. Many effects are not consistent across models, for example, the position of `Review` often behaves like a model-specific stylistic feature rather than a general predictor. Nonetheless, we observe the following patterns: (i) Correlations are stronger and more frequent in math reasoning than in general scientific reasoning; (ii) `Review-Chunk Fraction` shows weaker and more unstable association with accuracy, compared with `FSF`, suggesting that graph-level metrics are the more informative granularity; (iii) `Motivation Score` exhibits mixed, model-dependent correlations. This feature is intuitively important for human reasoning, as it gauges whether each action is taken with a clear purpose. For LRMs, however, it shows no consistent correlation with accuracy, suggesting their reasoning dynamics can differ from human patterns. (iv) For math reasoning, nearly all models show a positive association between `Reasoning Depth` and accuracy.

In addition to these metrics, we report analyses of other graph-based measures as well as entropy and progressiveness; see Appendices D and F. A model-level correlation analysis in Appendix F shows that models exhibit different generation styles, so comparing metrics *across* models may be biased. This supports our methodology: estimate correlations *within* each model, then seek patterns that replicate *across* models.

# D. Graph Metrics

The prompt for generating the reasoning graph:

> Parse the reasoning trace into a Graphviz diagram. Focus on these essentials:
>
> Node Rules:
> - One node per distinct reasoning step
> - 'fillcolor=lightblue': Successful reasoning steps
> - 'fillcolor=lightpink': Failed attempts
>
> Edge Rules:
> - Connect node A → node B if the information or insight from A is actually used to construct the reasoning in B; branch new attempts from their starting ancestor, not from dead ends.
>
> Requirements:
> - Use 'rankdir=TB'
> - Include ALL attempts (including failures), do not miss any steps in the reasoning.
> - ALWAYS start with a "problem statement" node
> - ALWAYS end with a "final answer" node
> - Do NOT reorder or reorganize the reasoning flow
>
> Generate complete Graphviz DOT code in dot blocks.

## D.1. Validation and Limitations of Graph Extraction

Graph extraction from free-form CoTs is necessarily lossy: long traces do not generally determine a unique graph. We therefore do not treat the extracted graph as a gold reconstruction of a latent reasoning graph. Instead, we use it as an operational representation for measuring stable aggregate structure, especially the fraction of abandoned reasoning steps. The syntactic validation is that all Claude-produced Graphviz files compiled successfully, but this only confirms that the DOT code is well formed. The main validation is functional: FSF is the strongest and most consistent predictor of correctness across models and datasets, remains positive under a different extractor model (Appendix D.4), outperforms the alternative graph metrics listed below, and supports the controlled editing intervention in Section 5.2.

The main extraction ambiguities we observed are structural rather than syntactic. First, a failed exploration can still partially inform a later successful direction, making the boundary between a side branch and the main path ambiguous. Second, node scope is not unique: the same local reasoning can be represented as several fine-grained nodes or one summarized node, especially when traces contain repeated algebraic manipulation or verification. Third, short-lived direction changes may be merged into nearby steps rather than represented as separate branches. These ambiguities affect exact node-by-node fidelity, but FSF is a ratio-level statistic and is therefore less sensitive to small differences in segmentation or local edge placement than metrics that depend on exact paths. We consequently interpret FSF as a robust structural summary, not as a claim of perfect graph reconstruction.

## D.2. Extra Graphical Metrics

A complete list of features we extract from the reasoning graph:

**Failed steps features**

`Failed-Step Fraction`: Proportion of nodes marked as failed steps, indicating the density of failed attempts in the reasoning process.

`Recovery Efficiency`: Average distance from failed nodes to successful nodes, measuring how quickly failed attempts can be corrected.

**Logical Flow Features**

`Branching Quality`: Fraction of decision points (nodes with multiple outputs) that lead to successful outcomes, assessing

the effectiveness of reasoning branches.

`Flow coherence`: Proportion of nodes that participate in paths connecting the problem statement to the final answer, measuring logical consistency.

**Structural Quality Features**

`Reasoning Depth`: Shortest path length from problem to answer node, representing the minimum logical steps required.

`Orphaned Steps`: Proportion of nodes with no incoming edges (excluding the problem node), measuring isolated reasoning steps.

**Information Utilization Features**

`Information Cascade`: Average number of downstream nodes reachable from each node, measuring information propagation potential.

`Cross Reference Density`: Proportion of nodes receiving input from multiple sources, indicating reasoning step validation.

**Path Features**

`Reasoning Efficiency`: Proportion of nodes involved in any path from problem to answer, measuring network utilization for reasoning.

`Shortest Path Coverage`: Fraction of total nodes on the shortest problem-to-answer path, indicating reasoning directness.

`Endpoint Reachability`: Proportion of nodes that can contribute to reaching the final answer.

**Error Analysis Features**

`Min Error Depth`: Minimum distance from problem node to any error node, indicating how early errors occur.

**Additional Structural Features**

`Total Steps`: Total number of nodes in the reasoning graph.

`Mean out Degree`: Average number of outgoing connections per node, measuring branching tendency.

`Max Failed Children`: Maximum number of failed nodes connected to any single node.

### D.3. Extra Graphical Results

Figure 9 reports correlation tests for the features above. Two observations stand out: (i) several features exhibit nontrivial correlations across many models, though their effects are markedly weaker than FSF; (ii) correlations are consistently significant for mathematical reasoning but are sparse for scientific reasoning, indicating limited generalization compared with FSF.

### D.4. Alignment across Extractor Models

Our reasoning graph extraction relies on Claude 3.7's ability to generate Graphviz code. To test whether FSF is specific to this extractor, we repeated graph extraction with GPT-4o on the same CoTs. We used the DeepSeek R1 and Claude 3.7 Chain-of-Thought (CoT) data from the HARP dataset and measured the agreement between the two extractors.

As shown in Table 4, we found a strong correlation for DeepSeek R1 ($r = 0.82$), which matches typical human expert agreement. The correlation for Claude 3.7 was moderate ($r = 0.67$). These positive correlations do not imply perfect graph recovery, but they indicate that the aggregate FSF signal is not an artifact of a single extractor model.

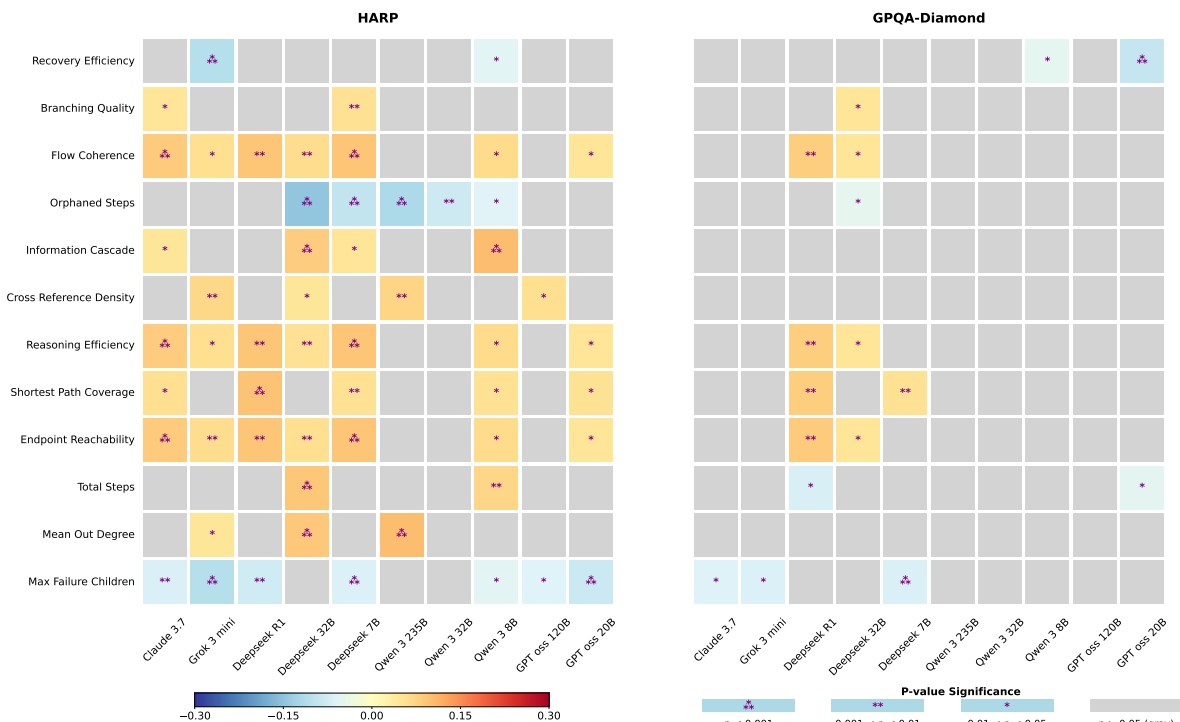

*Figure 9.* Correlation results for extra graphical metrics, computed on the full dataset. Again, correlations are shown with a color scale; non-significant cells ($p > 0.05$) are grayed out, and * denotes statistical significance. Overall, these metrics are weaker than FSF, with significant correlations appearing primarily in mathematical reasoning. We color * white or purple for visualization only.

*Table 4.* **Extractor Consistency Check.** We measure the correlation ($r$) between FSF values derived from two different graph extractors (Claude 3.7 and GPT-4o). Higher correlation indicates greater stability of the aggregate FSF measurement.

| CoTs | Deepseek R1 | Claude 3.7 |
|---|---|---|
| $r_{\text{Claude 3.7 vs GPT 4o}}$ | 0.82 | 0.67 |

## E. Intervention Details

### E.1. Test-Time Selection

Beyond the test-time selection results in Figure 5 (AIME-2025), we repeat the experiment on HARP. Specifically, we sample 180 questions (60 each from Levels 4, 5, and 6), disjoint from those used in our correlation analyses. For each question, we generate 64 CoTs and select the top-1 candidate under each metric. We generate 64 CoTs as in Appendix B, using four temperatures with 16 CoTs per temperature.

The results in Figure 10 show improvements for Claude 3.7, Grok 3 mini, and Deepseek R1 that mirror Figure 5. By contrast, Qwen exhibits anomalous behavior: selecting the generation with the smallest `length` or the smallest `Review Ratio` drives accuracy to 0. We hypothesize that this stems from train–evaluation contamination: these hard math problems (from past U.S. math olympiad contests) are likely included in Qwen's training (Reinforcement Learning from Verifiable Reward), leading to atypical selection dynamics. On contamination-minimized datasets (AIME 2025 and GPQA-Diamond), Qwen follows the same trend as the other models. Accordingly, we present the clean test-time selection results on these two datasets in the main paper, with AIME 2025 providing the most contamination-free evaluation.

### E.2. Controlled Editing of the CoT

In the intervention experiment, we need to identify where the failed branch begins and ends to remove it completely. We break it into several tasks, when Claude model extract the reasoning graph, we further ask it to (i) list each reasoning step and output the first 20 words of that step, and (ii) for each failed attempt step, mark the index at which the failed branch

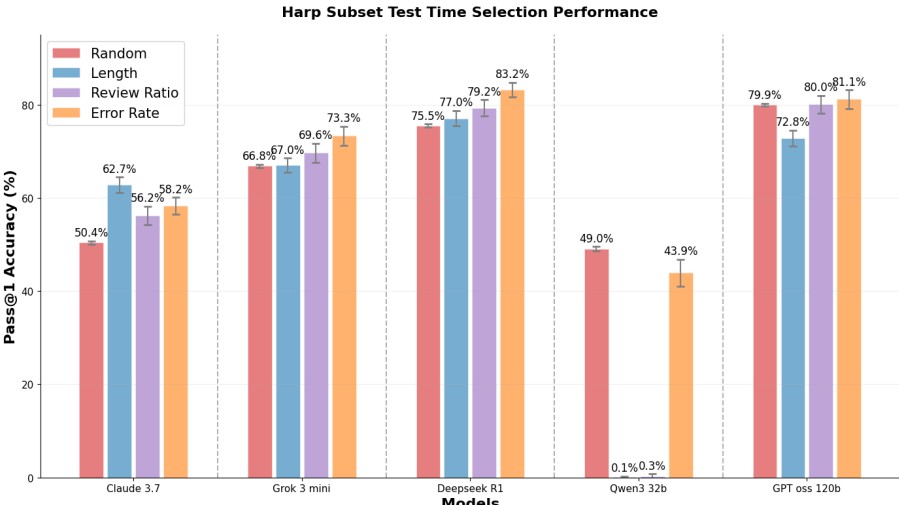

*Figure 10.* Pass@1 with test-time selection by `length`, `Review Ratio`, and FSF on HARP subset. Error bars show bootstrap standard deviations. Qwen exhibits weird results, likely because most evaluation questions appear in its RL training data. Excluding Qwen, FSF-based selection consistently identifies higher-quality generations at test time.

starts. (i) helps us build a mapping between the step in the graph to the sentences and paragraphs in the reasoning chain. We then align the returned quotations to the original CoT using n-gram matching (to tolerate minor misquotations). (ii) helps us to identify where to remove. We perform all of them together in one prompt, with the following prompt appended after the graph-extraction prompt.

The prompt used (to be appended after the graph-extraction prompt) is:

Additionally, provide a separate list with the exact format below:
List of nodes with first 20 words:
1. node id: "exact first 20 words of this reasoning step"
2. node id: "exact first 20 words of this reasoning step"
3. node id: "exact first 20 words of this reasoning step"
...

Requirements:
- Use numbered list format: "number. node id: "quoted text""
- Each entry must be on a single line
- Preserve exact formatting, punctuation, line breaks, and special characters from the original reasoning trace
- Use double quotes around the 20-word excerpts; the 20-word should be exactly the first 20 words of the reasoning step.
- Node IDs should match exactly with the DOT code node names
- This list should enable precise string matching back to the original reasoning trace

Example format:
1. problem statement: "Solve the following math problem efficiently and clearly. Please reason step by step, and put your final answer within $\boxed{answer}$."
2. analysis step: "First, let me understand what we're given. We have a triangle with specific angle measures and need to find the missing side length."
After these, for each failed attempts you have labeled as lightpink, tract the entire reasoning branch, also provide:

Branch Analysis:
1. node id, starts from node id "name", fails to current node id.

The definition: For each failed reasoning attempt (pink node), identify the most recent successful node (blue node) from which this failed path originally diverged, marking that successful node as the branch starting point where alternative reasoning paths split off. The next reasoning step after the current failed attempt should directly starts again from the node just before this branching starting point.

# F. Further Findings

## F.1. Progressiveness and Entropy

**Progressiveness and Entropy.** Motivated by recent work (Wu et al., 2025a), we evaluate how quickly a model converges to an answer (progressiveness) and how its answer entropy evolves along the CoT. The answer entropy measures how confident the model is with the answer. At multiple prefix truncations within each trace (0%, 25%, 50%, 75%), we append *I have thought long enough. Now let me conclude: the final answer is* to elicit a distribution over answers. We sample 8 continuations per truncation position and estimate confidence via the empirical answer entropy

$$H_t = -\sum_a \hat{p}_t(a) \log \hat{p}_t(a), \quad \text{where } \hat{p}_t(a) \text{ is the frequency of answer } a \text{ at checkpoint } t.$$

Compared with accuracy alone, the entropy captures the confidence and directly measures information accumulation. The actual information gain can be extracted with area under the curve, with normalization to $[0, 1]$ for the steps, `Progressiveness`: For reasoning trace $r = \{c_t\}_{t=1}^T$:

$$\texttt{Progressiveness}(r) = H_0 - \frac{1}{T}\sum_{t=1}^{T} H_t. \tag{2}$$

**Findings.** Figure 11 plots entropy and accuracy as functions of the truncation rate for CoTs on HARP, comparing Deepseek R1 and Qwen 3 8B. The x-axis denotes the truncation rate (fraction of characters removed from the end of the CoT): 0% = no truncation (original CoT), 95% = remove the last 95% of characters. Across models and difficulty levels, entropy

declines along the CoT with strikingly similar trajectories, and terminal entropy is low regardless of whether the final answer is correct or incorrect. In other words, models end up confident—even when wrong. Consequently, we do not include progressiveness and answer entropy in our correlation analyses.

In Figure 11, for each question we partition CoTs into a *short* half and a *long* half (by length) and report accuracy for each group across truncation rates. Across all difficulty levels, the *short* group attains higher accuracy, reinforcing that shorter CoTs correlate with higher accuracy.

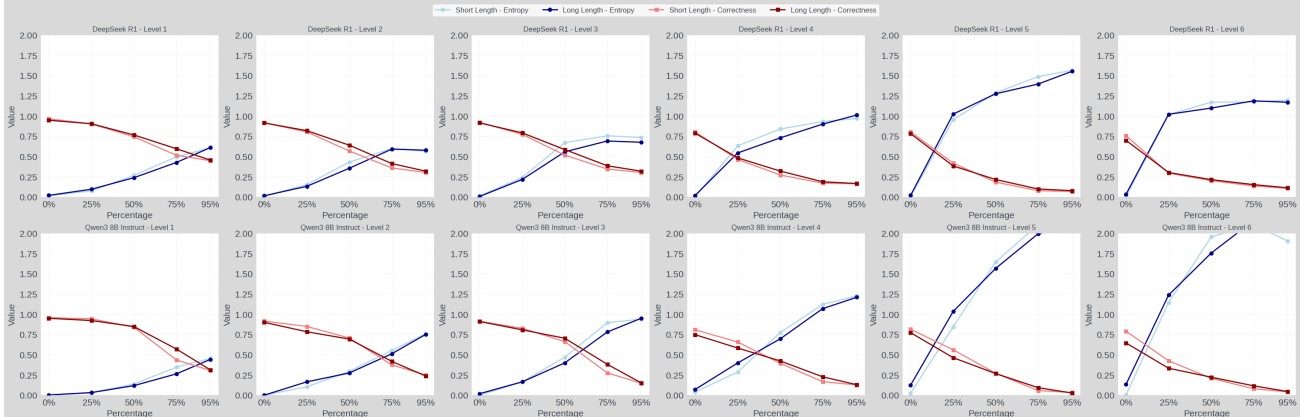

*Figure 11.* Impact of CoT truncation on answer entropy and correctness across difficulty strata. Within each question, we partition CoTs into *short* and *long* groups to compare length effects. The x-axis reports the truncation rate from the end of the CoT (e.g., 0 = no truncation; 0.5 = last half removed). Top: Deepseek R1; Bottom: Qwen 3 8B.

### F.2. How Different Models Behave

Finally, we are also interested in how different models have different behaviors on model level. Thus, we aggregate the average `Length`, `Review Ratio`, and `Failed-Step Fraction` for each model, and plot its distribution with model's accuracy. We present the results in Figure 12.

We do not observe a correlation pattern that holds uniformly across models. The clearest cross-model signal is `FSF` —especially on GPQA-Diamond—where models with lower `FSF` tend to achieve higher accuracy. The absence of universal trends is intuitive: output style strongly affects these metrics, particularly `length` and `Review Ratio`. Some models "over-verify" (Chen et al., 2024), inflating `length` and `Review Ratio` without necessarily lowering accuracy if the problem is ultimately solved. This further supports our decision not to pool CoTs across models. Instead, we take a more debiased analysis: we estimate correlations *within* each model and then look for patterns that replicate *across* models.

### F.3. Additional Dataset

To assess the generalizability of our observations to different reasoning domains, we investigated coding behaviors and additional reasoning scenarios. To this end, we selected 80 questions from MMLU-pro (Wang et al., 2024), comprising 40 from the Computer Science category and 40 from the Engineering category. We followed the same conditional correlation setup described in Section 4. The correlation results are visualized in Figure 13.

In the Computer Science category, we observe behavioral differences across models. `Length` is a significant feature only for the Qwen model family, while `FSF` is significant for other models, including Claude 3.7 and Deepseeks. Conversely, in the Engineering category, both `Length` and `FSF` generally exhibit a negative effect—consistent with our observations for math and general reasoning—with `FSF` showing significance across a wider range of models. These results underscore the importance of `FSF` for both computer science and engineering tasks.

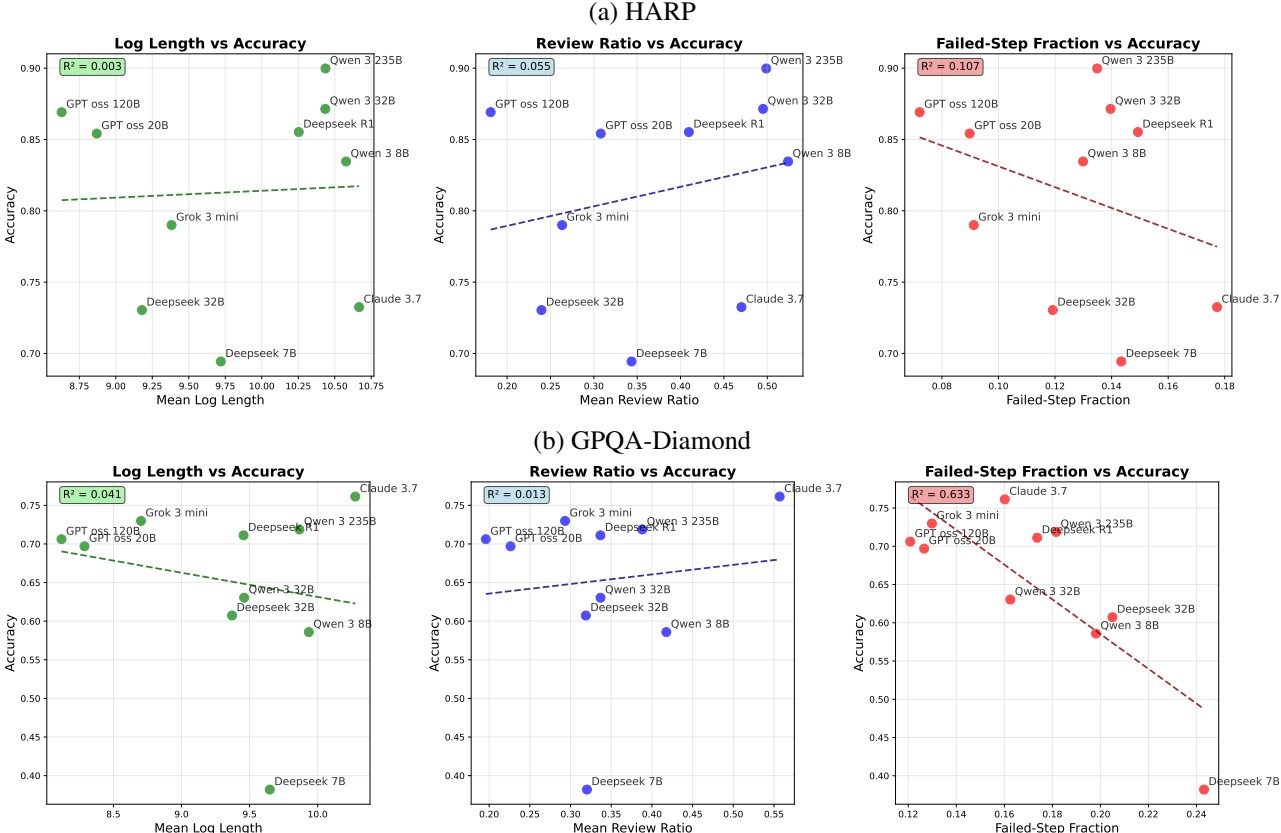

*Figure 12.* Model-level relationship between accuracy and average behavior (`length`, `Review Ratio`, FSF). Top: HARP; Bottom: GPQA-Diamond. Overall, we do not observe consistent cross-model patterns: these features are largely model-specific. Though FSF shows some correlation across models, especially for GPQA-Diamond.

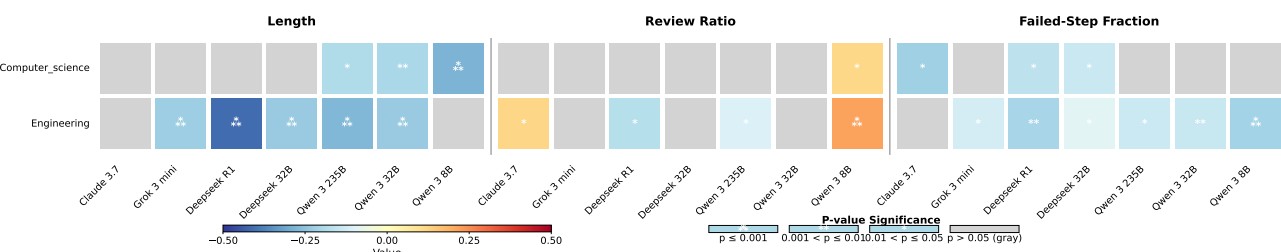

*Figure 13.* Conditional correlations computed on the Computer Science and Engineering class of MMLU-pro. Correlations are shown with a color scale; non-significant cells ($p > 0.05$) are grayed out, and * denotes statistical significance (see the legend). We color * white or purple for visualization only. More colored cells indicate broader prevalence of correlation; darker colors denote stronger correlations.

