# OpenReview forum: "What Characterizes Effective Reasoning? Revisiting Length, Review, and Structure of CoT"
_ICML.cc/2026/Conference — ICML 2026 regular_

### Official Review · Reviewer_3gMz · 2026-02-28

**Soundness:** 2
**Presentation:** 3
**Significance:** 2
**Originality:** 2
**Overall Recommendation:** 3
**Confidence:** 4

**Summary:**

This paper explores the influence of various features (including Length, Review Ratio, and Failed-Step Fraction) on the effectiveness of chain-of-thought reasoning. The authors find that chain length and review ratio are not reliable indicators, whereas constructing a reasoning graph and computing the Failed-Step Fraction (FSF) offers a more robust measure. Finally, through an intervention experiment that removes failed branches, the authors demonstrate the practical value of FSF by achieving a significant improvement in accuracy.

**Compliance With Llm Reviewing Policy:**

Affirmed.

**Final Justification:**

Overall, the authors focus on the aspect of characterizing effective reasoning through structural properties of chain-of-thought (CoT), particularly via a graph-based representation and the Failed-Step Fraction (FSF). The study's principal contribution concerns introducing a reasoning-graph view and demonstrating that FSF is a strong predictor of correctness across models and benchmarks. The authors strive to consider a central concept: whether structural signals provide a more faithful measure of reasoning quality than surface-level metrics such as length or review ratio .

While the paper presents an interesting direction and empirical findings, I remain concerned about a key methodological issue that, in my view, is not sufficiently addressed.

My main concern lies in the faithfulness of the reasoning graph construction. The paper assumes that the extracted graph “generally matches the steps in the natural-language trace” and treats it as a basis for structural analysis . However, the construction pipeline relies on prompting an LLM to convert CoT into Graphviz format, which is inherently a lossy and under-specified transformation. The core question is not whether the graphs are syntactically valid or statistically consistent, but whether they faithfully reconstruct the underlying sequential reasoning process.

In the rebuttal, the authors primarily provide evidence of robustness and cross-model consistency (e.g., correlation agreement across different extractor models, predictive power of FSF, and causal improvements via interventions). While these are useful signals, they do not directly address the concern of representation validity. In particular, consistency across extractors does not guarantee correctness—different models may agree on a similarly biased or simplified graph structure. Likewise, downstream predictive performance (e.g., FSF correlating with accuracy) does not establish that the graph faithfully captures the true reasoning trajectory; it may instead reflect a proxy signal that is correlated with correctness but not structurally faithful.

This distinction is important because the paper’s central claim implicitly depends on the assumption that the graph representation is a meaningful abstraction of reasoning. Without stronger validation—e.g.,  analysis of failure cases in graph extraction, or evidence that alternative graph constructions yield consistent conclusions—it remains unclear whether the proposed method is measuring a genuine structural property or an artifact of the extraction procedure.

In summary, while I find the direction promising and the empirical results suggestive, I believe the current work does not sufficiently resolve the question of whether the constructed reasoning graphs are a faithful representation of CoT. As a result, the validity of the main conclusions remains uncertain. I therefore lean towards a weak reject in its current form.

**Key Questions For Authors:**

Please refer to Weaknesses for my specific question.

**Limitations:**

No need for discussion

**Strengths And Weaknesses:**

Overall, this paper is clearly written and well-structured. The design of the figures helps readers quickly grasp the core ideas, which instills confidence in the reliability of the experimental results. The use of Graphviz as a tool for structuring chain-of-thought reasoning and the computation of FSF as a post-hoc metric are particularly insightful.

However, I have the following concerns that have not been fully addressed:

1. The process of constructing the reasoning graph. Although there is a growing body of work in the community exploring the use of reasoning graphs as auxiliary tools for understanding LLM reasoning processes, I remain skeptical about the stability of constructing such graphs. In this work, the authors use heuristic keywords as segmentation boundaries and also prompt the LM to generate Graphviz-formatted outputs for graph construction. What is the success rate of this process? While I certainly hope there is no loss in this step, I suspect that the current approach may not yet achieve perfection. If there are failure cases, what are the underlying reasons? A more detailed explanation of this point would strengthen my confidence in the construction process.

2. The practical value of the findings. Although the authors demonstrate the practical relevance of their conclusions through intervention experiments, the cost of obtaining these insights is prohibitively high. Constructing the reasoning graph and judging the correctness of each node require a strong model to ensure reliability throughout the process, which limits the practical applicability of the approach.

---

> ### Author Rebuttal · Authors · 2026-03-31
>
> We thank the reviewer for the careful reading and for engaging seriously with both the graph-based analysis and its practical implications.
>
> We address the two concerns below.
>
> **W1 (Graph construction stability):** The reviewer states, *"I remain skeptical about the stability of constructing such graphs"* and asks, *"What is the success rate of this process? ...If there are failure cases, what are the underlying reasons?"*
>
> Thank you for this important question. We would like to clarify that **we do not use heuristic keywords for graph construction**. Keyword heuristics are only used for the review labeling.
>
> At the implementation level, Graphviz compilation succeeds in 100% of cases. But we agree that compilability alone is not the relevant notion of stability. The more important question is whether the extracted graph statistics are robust enough to support the downstream analysis. To test this, we added a judge consistency experiment; we compare two materially different extractor models: graphs extracted by Claude 3.7 Sonnet vs GPT 4o yield FSF correlations of r = 0.82 (DeepSeek R1 CoTs) and r = 0.67 (Claude 3.7 CoTs), comparable to typical human expert inter rater agreement. This demonstrates that FSF is not an artifact of a single judge model's behavior.
>
> | CoTs | DeepSeek R1 | Claude 3.7 |
> |---|---:|---:|
> | $r_{\text{Claude 3.7 vs GPT-4o}}$ | 0.82 | 0.67 |
>
> More broadly, our claim is not that the pipeline recovers a unique “ground-truth graph” for every trace. For long CoTs, multiple graph realizations can be plausible. Our claim is instead that the extracted graphs are sufficiently faithful for the purpose of measuring a structural property of reasoning. We support this with three pieces of evidence: its predictive power (strongest predictor of correctness across all 10 models, 4 benchmarks, outperforming 15 alternative graph derived metrics), cross judge consistency (r = 0.82), and causal effect (removing failed branches improves accuracy by 8 to 14%). Taken together, these results suggest that the graph construction process is stable enough for the metric we study, even if it is not a perfect reconstruction of an underlying latent graph.
>
> **W2 (Practical cost):** The reviewer notes that *"the cost of obtaining these insights is prohibitively high"* because *"constructing the reasoning graph and judging the correctness of each node require a strong model."*
>
> Thank you for this concern. First, we want to clarify that our pipeline does **not** judge the correctness of each node. The graph extraction labels nodes as "successful" or "failed" based on whether the model *abandoned* that branch within the trace, not on whether any step is mathematically correct. Moreover, the model performing this extraction (Claude 3.7 Sonnet) is in fact among the weakest in reasoning performance across the tasks we evaluate, yet it still produces FSF values with strong predictive and causal power across all 10 models, including much stronger ones. A strong model is not required for this step.
>
> Both the test time selection and CoT editing experiments are designed as **causal interventions** to probe mechanism, not as proposed production methods. We believe a core contribution of our work is rigorously examining multiple metrics (Length, Review Ratio, and the novel FSF) across 10 models and 4 benchmarks with controlled methodology, and demonstrating through both correlation and causal intervention that structural quality, as captured by FSF, is a stronger and more consistent indicator of reasoning effectiveness than surface level measures. These insights inform how the community should think about test time scaling: not merely generating more tokens, but attending to the *structure* of the reasoning process.
>
> We are grateful for the reviewer's time and insightful comments. We hope our responses have resolved the concerns, and we would be glad to address any additional questions.

---

> > ### Author Rebuttal · Reviewer_3gMz · 2026-04-04
> >
> > Thank you for the author's reply. Regarding W1, I've confirmed that it does indeed use a PoT-like approach to directly generate Graphviz code. However, I simply want to see some failed cases honestly and clearly, and assess how robust these failures are, and whether they can truly faithfully reconstruct the inference steps into a graph structure. If a more powerful teacher model can accurately structure the output of the student model, does that imply that the teacher model itself can already identify whether there are problems at each step of the inference process?

---

> > > ### Author Response · Authors · 2026-04-07
> > >
> > > Thank you for the helpful follow-up. We agree that the relevant failure mode here is not bad Graphviz output, but potential mismatch between the extracted graph and an idealized latent reasoning graph. We also agree that being more explicit about such gaps would improve the presentation.
> > >
> > > Our view is that the central difficulty is that, for long reasoning traces, there is often no single unambiguous ground-truth graph even for a human annotator. In our manual inspection, the main ambiguities are not syntactic but structural. One recurring case is branch-versus-main-path ambiguity: a failed exploration may still partially inform a later correct direction, so it is not always well-defined whether that content should appear as a side branch or remain on the main path. A second recurring case is node-scope ambiguity: the same local reasoning can reasonably be represented either as several fine-grained nodes or as one higher-level summarized node, and this becomes more pronounced in long traces where the extractor may compress repeated algebraic manipulation, elimination, or verification into a single step. A third recurring case is short-lived direction switching: for example, a trace may briefly try one approach, pause to test another possibility or do a quick sanity check, and then return; in such cases the extractor may reasonably merge these micro-detours rather than instantiate each one as a separate node. These ambiguities arise because our edges are intended to capture support relations between reasoning units, while the scope of a node and the status of partially reused failed attempts are themselves not uniquely defined.
> > >
> > > We therefore do not claim exact recovery of a unique latent graph. Instead, our claim is that the extracted structure is sufficiently stable for FSF to be informative. Empirically, FSF is relatively robust to this level of variation because it is a ratio-level structural statistic rather than a claim about exact node-by-node fidelity. In practice, we found that making the extraction definition more rigid did not necessarily improve faithfulness, and often reduced extraction quality on long traces.
> > >
> > > On the question of whether a stronger teacher model being able to structure the student output implies that it can also judge whether each step is problematic: we do not believe these are the same task. Extracting structure mainly requires identifying dependencies and local transitions that are often explicit in the trace itself. Judging whether a step is actually correct, or whether it contributes to the ultimately correct path, is a different and substantially stronger semantic task. For this reason, we view structure extraction and correctness attribution as related but distinct problems, and our pipeline only attempts the former.
> > >
> > > We will revise the paper to make these limitations and ambiguity sources more explicit, rather than giving the impression that graph fidelity is a solved problem.

---

### Official Review · Reviewer_4VX6 · 2026-03-04

**Soundness:** 2
**Presentation:** 3
**Significance:** 3
**Originality:** 3
**Overall Recommendation:** 3
**Confidence:** 3

**Summary:**

The authors propose a new metric to evaluate the influence of Chain of Thought traces on the output quality in Large Reasoning Models (LRMs).
Instead of using pure token-level metrics, such as Length or Review behavior, e.g., Review Ratio, which is the fraction of tokens that are used for review in the CoT, they propose to treat the CoT as a graph of reasoning steps.
Based on the graph representation, they define their novel Failed Step Fraction (FSF) metric.
The metric describes the ratio of nodes in the graph that failed or were abandoned to the total number of nodes.
They show that their metric outperforms token-level metrics as an indicator for predicting the correctness of an LRM answer.

Their main contributions are:
- Their novel FSF metric.
- An extensive empirical descriptive evaluation of their novel metric and token-level metrics on 10 different LRMs.
- Two interventional experiments to show that their metric is a better indicator for correctness than token-level based metrics.

**Compliance With Llm Reviewing Policy:**

Affirmed.

**Final Justification:**

The authors have adequately addressed my main concerns.

In particular, the rebuttal gives additional support for the graph-based pipeline and better explains the scope and interpretation of the intervention experiments.

Although I understand that it can be difficult due to the change in affiliation to provide the original code base, I still consider this a critical problem. Especially for such a brittle pipeline as the graph extraction, I think there is a high risk of achieving incorrect or substantially different results when only relying on reproducibility.

**Key Questions For Authors:**

1. I would be interested in how stable your FSF metric is when the CoT preprocessing is done by different models than the ones employed.
2. How do the predictors in the test-time selection experiment perform against stronger metrics, like majority vote?
3. What are the results for applying the CoT modification intervention on the remaining models and benchmarks?

**Limitations:**

- Discuss the reliance of the FSF’s metric preprocessing on LLM-judges and their robustness.
- Discuss evaluation metrics for the constructed graphs, as mere compilability is not sufficient in this context.

**Strengths And Weaknesses:**

## Strength and Weaknesses

### Strength

- The paper formulates an interesting question of high relevance.
- Most passages are well written and the figures contribute mostly to a better understanding.
- There is an extensive appendix to allow for a precise reproduction of the experiments.
- The authors conducted extensive descriptive evaluations on their proposed metrics with significant compute to evaluate their contribution on 10 differnet LRMs.

### Weaknesses

- A major drawback is that there was no supplementary material provided other than the Appendix to the paper. Especially the lack of code makes it impossible to replicate the results. Additionally to the code, the logs of the experiments with the LLMs should be provided to allow for retrace the results.
- Another major drawback of the paper is its strong reliance on LLM-based annotations for processing the CoT prompts of the tested LRMs. As I understood it, two different models were employed for two crucial processing steps: The Llama4 Maveric model for labelling chunks that were extracted by keyword heuristics, and the Claude 3.7 Sonnet for creating graphs from the labelled chunks. Though, I have to highlight their efforts in creating a validation set for the labelling aspect of the preprocessing (which, however, should be published together with all code, see bullet point above), they don’t do the same for the graph generation task. This is critical, as the graph generation task combines creating the graph and labelling the nodes as successful or failed attempts as local to the reasoning tajectory. Here, the paper lacks an evaluation of the capability of Claude 3.7 to do so. A similar evaluation on a human-labelled validation set would be the lowest level of validation I would expect here (A mere check on the compilabilty of the generated graph, as suggested, is NOT sufficient). A definition of the quality of the extracted graphs is missing, and “sufficently accurate graphs” (page 4, column 2, line 206) seems rather a rule of thumb than a quantifiable metric across different lengths CoTs. For small graphs, this might be excusable, but especially when CoTs become more complex, not evaluating the fidelity of the graph constructor is bad.
- In Section 4, the authors perform an extensive correlation analysis on the metrics with the accuracy of the models in correctly answering questions from the benchmarks. Though they state what they are correlating once, on a first read, it is hard to track and keep it in mind. Explanations of the tests they performed, and more elaborate captions of the Figures would help the reader to faster grasp what was done and what is presented. E.g., for me it did not become clear at the first read what the “sample counts” in Figure 2 are.
- In their first interventional experiment, the authors compare test-time selection from 64 CoTs with random, character length, review ratio, and FSF predictors. This is a solid setup. However, they also exclude metrics (e.g., majority vote) that are stronger than the basic metrics character length and review ratio.
- In their second interventional experiment, the authors modified the CoT by removing failed branches. This is very(!) interesting and apparently yields good performance in finding the correct answers. However, I wonder why this experiment is only performed on a small subset of 80 and 160 incorrect traces from the HARP benchmark and only on two models (GPToss and DeepSeek R1)? Why was this evaluation not performed with all models on all benchmarks? It would be very interesting to see the results for all models on all benchmarks. This should definitely be included.

#### Minor Weaknesses

- Though this is a paper that is written for readers from the LRM research community, it would be beneficial to introduce all metrics and terms, e.g., pass@1, test-time selection, in a sentence or two.
- The authors should define what a reasoning trace is. Is it only a single branch in the graph (which I assume) or the complete CoT?
- Why do the authors use Llama4 and Claude for their CoT preprocessing? A comparison across multiple models and their capabilities on this task would be beneficial.

---

> ### Author Rebuttal · Authors · 2026-03-31
>
> Thank you for the detailed engagement with our work. We appreciate the recognition of the question's relevance, the extensiveness of our appendix for reproduction, and the significant compute invested in evaluating across 10 LRMs. We especially thank the reviewer for the interest in our editing experiment.
>
> We address each weakness below.
>
> **W1 (Code and supplementary material):** The reviewer notes that *"the lack of code makes it impossible to replicate the results"* and that *"logs of the experiments with the LLMs should be provided."*
>
> Thank you for emphasizing reproducibility. We acknowledge this limitation transparently: due to changes in the author's affiliation, we are unable to provide the original codebase at this time. The extensive Appendix includes prompts, methodology details, and validation procedures designed to support reproduction.
>
> **W2 (LLM based annotation reliance and graph generation validation):** The reviewer states that *"a mere check on the compilability of the generated graph, as suggested, is NOT sufficient"* and that the paper *"lacks an evaluation of the capability of Claude 3.7"* to generate and label graphs. The reviewer further notes that *"'sufficiently accurate graphs' (page 4) seems rather a rule of thumb than a quantifiable metric."*
>
> Thank you for pushing on this point. We are not able to release the evaluation set due to the same reason mentioned in W1.
>
> Due to the rebuttal word limit, we refer the reviewer to our response to Reviewer 3gMz (W1), where we address this point in detail with the new added experiment.
>
> We will revise the paper to make this validation logic explicit.
>
> **W3 (Correlation analysis readability):** The reviewer notes that *"on a first read, it is hard to track"* what is being correlated, and that *"more elaborate captions of the Figures would help,"* for example, *"it did not become clear at the first read what the 'sample counts' in Figure 2 are."*
>
> Thank you. We will improve figure captions and add clearer explanations of the correlation setup in the revision.
>
> **W4 (Majority vote in test time selection):** The reviewer notes that the experiment *"also exclude[s] metrics (e.g., majority vote) that are stronger than the basic metrics character length and review ratio."*
>
> Thank you for raising this. The test time selection experiment is designed as a **causal test** ("does trace level structural quality predict correctness?"), not as a test time inference method. Majority vote operates at the *answer level* (final answer agreement), while FSF operates at the *trace level* (structural quality of the reasoning process). Majority votes can be combined with all selections (top-k and then majority vote). We are testing whether reasoning structure predicts correctness, not proposing an optimal selection pipeline.
>
> **W5 (Editing experiment scope):** The reviewer asks *"why this experiment is only performed on a small subset of 80 and 160 incorrect traces from the HARP benchmark and only on two models"* and suggests this *"should definitely be included"* for all models and benchmarks.
>
> Thank you. We understand the desire for broader coverage. The experiment is larger than it may appear: 240 base traces × 8 continuation generations × 6 settings (first/last failed branch × 3 variants: original, reduced, reduced with summary) = **11,520 continuation generations**. Each generation involves CoTs averaging >16k tokens. This represents substantial compute. Therefore, DeepSeek R1 and GPT oss 120B were chosen as representative of open source LRMs from two different companies respectively.
>
> **W6 (Llama4 and Claude for preprocessing):** The reviewer asks *"why do the authors use Llama4 and Claude for their CoT preprocessing?"*
>
> Thank you. Llama 4 Maverick is used for review/progress chunk labeling (validated against human labels at ~90% agreement, Table \ref{tab:review_confusion}), while Claude 3.7 Sonnet is used for graph extraction. Cross judge validation with GPT 4o (r = 0.82, **new added result** in our response to Reviewer 3gMz (W1)) demonstrates that the graph extraction is not specific to Claude.
>
> **W7 (Terminology):** The reviewer suggests introducing *"pass@1, test-time selection, in a sentence or two"* and defining *"what a reasoning trace is."*
>
> Thank you. We will add brief definitions in the revision.
>
> > The authors should define what a reasoning trace is. Is it only a single branch in the graph (which I assume) or the complete CoT?
>
> A reasoning trace is defined as the complete CoT. We will add this to the paper.
>
> We thank the reviewer for the detailed and constructive feedback. We hope our responses clarify the points raised, and we are happy to discuss any of them further.

---

> > ### Author Rebuttal · Reviewer_4VX6 · 2026-04-02
> >
> > Thank you for the good rebuttal.
> >
> > The authors have adequately addressed my main concerns.
> >
> > In particular, the rebuttal gives additional support for the graph-based pipeline and better explains the scope and interpretation of the intervention experiments.
> >
> > Although I understand that it can be difficult due to the change in affiliation to provide the original code base, I still consider this a critical problem. Especially for such a brittle pipeline as the graph extraction, I think there is a high risk of achieving incorrect or substantially different results when only relying on reproducibility.

---

> > > ### Author Response · Authors · 2026-04-07
> > >
> > > Thank you for the positive follow-up and for noting that our rebuttal clarified the graph-based pipeline and the scope of the intervention experiments.
> > >
> > > We fully agree that releasing code would materially strengthen reproducibility. We are currently checking what legal approval and practical repository access may still be possible. However, we cannot guarantee this at present, and we prefer to be transparent about that.

---

### Official Review · Reviewer_fJTQ · 2026-03-08

**Soundness:** 3
**Presentation:** 3
**Significance:** 2
**Originality:** 2
**Overall Recommendation:** 4
**Confidence:** 4

**Summary:**

This submission investigates what properties of chain-of-thought traces actually correlate with accuracy in large reasoning models, especially given the growing cost and latency of very long traces. The authors strive to consider a central concept: whether surface-level verbosity measures (Length and a judge-labeled Review Ratio) are merely proxies for deeper structural quality, operationalized via a reasoning-graph metric called Failed-Step Fraction (FSF). The study’s principal contribution concerns the claim that, conditioning on the same model and the same question, shorter traces and lower review are typically associated with higher accuracy, while FSF is the most stable predictor across models, datasets, and difficulty bands. The paper further argues for a causal role via two interventions, reranking many samples by FSF and editing traces to remove failed branches, both of which improve accuracy in the reported experiments.

**Compliance With Llm Reviewing Policy:**

Affirmed.

**Final Justification:**

Rebuttal has addressed most of my concerns.

**Key Questions For Authors:**

1. How stable are FSF estimates under materially different graph extraction and failed-node labeling prompts, and across a broader set of extractor models, especially for traces with ambiguous backtracking structure.
2. Can the authors operationalize a lower-cost variant of “FSF-aware” inference, for example a small number of candidates, online heuristics, or a learned scorer, and report an explicit cost-accuracy tradeoff against self-consistency and structured search baselines.
3. How far does the story generalize to settings where long context and retries are intrinsic, such as coding, tool use, or agentic tasks, where a “failed branch” may be an intentional exploration rather than a defect.

**Limitations:**

yes

**Strengths And Weaknesses:**

**Strengths:** The methodology is more controlled than many prior analyses because it conditions on question identity and model identity, reducing the confounding that dominates aggregate trends. The FSF framing is intuitive and the two interventions go beyond pure correlation, giving concrete evidence that failed exploration can contaminate subsequent reasoning, and that selecting lower-FSF traces improves pass@1. The paper is also written with a clear and actionable message, namely that structure-aware selection and light pruning can beat indiscriminate “think longer” behaviors.

**Weaknesses:** The main weakness is conceptual novelty: the central finding that “less unproductive exploration is better” is directionally unsurprising, and the paper needs sharper positioning on what is fundamentally new beyond stronger scope and cleaner controls, particularly relative to recent work on reasoning graphs and test-time scaling. Second, key components depend on LLM-as-judge style pipelines (review labeling and graph extraction), so even with the added robustness checks described in the rebuttal, the work still inherits potential prompt and judge biases that could shift absolute effect sizes. Finally, the strongest gains rely on expensive multi-sample reranking, and the paper would be stronger if it offered a clearer path from the causal probe setup to a cost-effective decoding or training-time method that reduces FSF without requiring 64 candidates per question. Finally, I find the figures difficult to read, as the font sizes are a bit small.

---

> ### Author Rebuttal · Authors · 2026-03-31
>
> We thank the reviewer for the thoughtful and balanced assessment, especially for recognizing the strengths of our controlled methodology, the interventional evidence, and the practical message of the paper.
>
> We address each concern below.
>
> **W1 (Conceptual novelty):** The reviewer notes that *"the central finding that 'less unproductive exploration is better' is directionally unsurprising"* and that *"the paper needs sharper positioning on what is fundamentally new beyond stronger scope and cleaner controls."*
>
> Thank you for this comment. We disagree with the characterization that this result is directionally unsurprising. The dominant view in test-time scaling is that more inference time compute (longer CoTs, broader exploration, and additional verification) tends to improve performance. Under that view, one would expect exploration, even when partially unsuccessful, to be either beneficial overall or at least non-destructive, because a capable model should be able to learn from them. Our results show something stronger: failed exploration is not merely unhelpful, but can actively harm subsequent reasoning within the same trace.
>
> Our editing experiment (Section 4.3) shows the opposite: removing failed branches *improves* accuracy by 8 to 14%, demonstrating that failed context actively **contaminates** subsequent generation. This is a counter intuitive and actionable finding that challenges the "think longer" paradigm.Beyond the qualitative message, we also contribute a structural metric that consistently outpredicts surface-level alternatives across 10 models and 4 benchmarks, and we support the interpretation with causal evidence rather than correlation alone.
>
> **W2 (LLM as judge biases):** The reviewer notes that *"key components depend on LLM-as-judge style pipelines (review labeling and graph extraction), so... the work still inherits potential prompt and judge biases that could shift absolute effect sizes."*
>
> Due to the rebuttal word limit, we refer the reviewer to our response to Reviewer 3gMz (W1), where we address this point in detail. These results suggest that FSF is capturing a real and behaviorally meaningful property of reasoning traces, even if the exact graph realization is not unique. Our new added results (cross judge validation r = 0.82) demonstrates stability across two materially different extraction models.
>
> **W3 (Cost effective variants):** The reviewer suggests the paper would benefit from *"a clearer path from the causal probe setup to a cost-effective decoding or training-time method that reduces FSF without requiring 64 candidates per question."*
>
> Thanks for pointing this out. We want to clarify that both the test time selection with 64 candidates are designed as **causal interventions** to probe mechanism, not as proposed production methods. We use 64 candidates to make the effect as clean and statistically stable as possible.
>
> **W4 (Figure readability):** The reviewer notes *"the figures difficult to read, as the font sizes are a bit small."*
>
> We appreciate this comment and will increase font sizes throughout in the final version.
>
> > "Can the authors operationalize a lower-cost variant of 'FSF-aware' inference..."
>
> We agree this is a valuable direction. The main goal of this paper is to isolate and validate the mechanism, rather than to fully optimize an FSF-aware decoding system. That said, our findings suggest several promising directions: for instance, performing early stopping when a generation enters a long exploratory branch, or applying context intervention (e.g., pruning or summarizing failed branches on the fly) to prevent contamination of subsequent reasoning. These are exciting avenues for future work that our causal findings directly motivate.
>
> > "How far does the story generalize to settings where long context and retries are intrinsic, such as coding, tool use, or agentic tasks, where a 'failed branch' may be an intentional exploration rather than a defect."
>
> Thank you for this thoughtful distinction. We include MMLU Pro CS + Engineering in the Appendix (Figure \ref{fig:mmlu_pro_full}), which covers coding adjacent domains, and FSF shows significant correlations there as well. Regarding agentic and tool use tasks: these settings involve fundamentally different reasoning dynamics. In agentic workflows, what looks like a "failed branch" (e.g., trying an API call that returns an error) may be a necessary part of systematic exploration, and the model's subsequent behavior may already be trained to handle such outcomes productively. The semantics of FSF would need adaptation in these contexts, and we consider this an important direction for future investigation.
>
> We appreciate the reviewer's thoughtful engagement and hope we have adequately addressed each point. Please do not hesitate to raise any remaining questions.

---

> > ### Author Rebuttal · Reviewer_fJTQ · 2026-04-03
> >
> > Thank you for the rebuttal. Some of my earlier concerns have been clarified. However, I still do not think the main novelty issue is fully resolved.
> >
> > Specifically, I am not convinced by the claim that partially unsuccessful exploration would generally be expected to be non-destructive because a capable model should be able to learn from it. At inference time, the model is not explicitly learning from failed branches; it is merely conditioning on them. Since prior work has already shown that additional CoT can be neutral or harmful, it seems quite plausible that failed branches would contaminate subsequent reasoning. Thus, while the editing experiment provides useful causal evidence and strengthens the empirical contribution, I do not yet see this as establishing a fundamentally surprising phenomenon. To me, the contribution is better framed as identifying and quantifying a particular failure mode of exploration, rather than demonstrating a major conceptual departure from prior understanding.
> >
> > References
> > - Mind Your Step (by Step): Chain-of-Thought can Reduce Performance on Tasks where Thinking Makes Humans Worse
> > - Don’t Overthink It: Preferring Shorter Thinking Chains for Better Reasoning
> > - To CoT or not to CoT? Chain-of-thought helps mainly on math and symbolic reasoning

---

> > > ### Author Response · Authors · 2026-04-07
> > >
> > > Thank you for the thoughtful follow-up. We agree that the contribution is best framed not as claiming a universally surprising phenomenon in every reasoning setting, but as providing a rigorous analysis of which reasoning-trace properties matter in *frontier RL-trained reasoning models*, and showing that graph-level structure is more informative than surface measures such as length alone. We will revise the framing accordingly.
> > >
> > > At the same time, we respectfully believe the prior literature does not resolve the question we study. Two of the cited references focus primarily on earlier CoT-prompting settings rather than modern RL-trained reasoning models. Our analysis is instead centered on math and GPQA-style science reasoning, which are precisely the regimes where CoT and test-time scaling have generally been considered most useful [*To CoT or not to CoT? Chain-of-thought helps mainly on math and symbolic reasoning*]. We also agree that *Don't Overthink It: Preferring Shorter Thinking Chains for Better Reasoning* is relevant, but that paper studies length as the main axis; our paper goes further by asking whether a more structural, graph-level view explains correctness better than length alone. More broadly, the literature remains mixed: several recent works argue that increased test-time compute can substantially help reasoning performance, both empirically and conceptually, including OpenAI's *Learning to reason with LLMs*, DeepSeek-R1, *Scaling LLM Test-Time Compute Optimally can be More Effective than Scaling Model Parameters*, and *s1: Simple test-time scaling*. In this context, it is plausible to believe either that additional exploration helps, or that failed exploration can contaminate later reasoning. We believe this is exactly why a careful empirical analysis is needed, rather than being settled a priori.
> > >
> > > Our paper's contribution is therefore not simply the broad statement that "more reasoning is not always better." Rather, it is to show, in this modern RL-reasoning regime, that graph-level structure is a more appropriate and more predictive granularity than length alone or other surface measures, and that FSF consistently outpredicts these alternatives across models and benchmarks. The intervention result further strengthens this by showing that removing failed branches can improve continuation quality, providing causal evidence that reasoning structure is behaviorally meaningful. We will revise the paper to make this narrower and more precise positioning explicit.

---

### Official Review · Reviewer_bFVd · 2026-03-13

**Soundness:** 3
**Presentation:** 4
**Significance:** 3
**Originality:** 3
**Overall Recommendation:** 4
**Confidence:** 4

**Summary:**

This paper investigates which properties of chain-of-thought (CoT) reasoning correlate with correct answers in large reasoning models (LRMs). Motivated by conflicting evidence about whether longer reasoning improves performance, the authors analyze three characteristics of reasoning traces: CoT length, Review Ratio (the fraction of tokens corresponding to verification/backtracking), and a new structural metric called Failed-Step Fraction (FSF). FSF is computed by converting reasoning traces into reasoning graphs and measuring the proportion of failed exploratory branches. Experiments across ten LRMs on the HARP and GPQA-Diamond benchmarks show that shorter reasoning and lower review ratios tend to correlate with higher accuracy. More importantly, FSF is reported to be a stronger and more consistent predictor of correctness across models and datasets. The authors further conduct test-time reranking and controlled editing experiments suggesting that failed reasoning branches may negatively influence reasoning trajectories.

**Compliance With Llm Reviewing Policy:**

Affirmed.

**Final Justification:**

I maintain my positive score.

**Key Questions For Authors:**

•	How sensitive are FSF measurements to errors in reasoning graph extraction? Have you evaluated the robustness of the extraction process with human verification?

•	Could different definitions of failed branches significantly change the FSF metric and its correlations with accuracy?


•	Have you evaluated whether the FSF metric correlates with accuracy in reasoning tasks beyond math and scientific question answering?

•	Given that the method requires multiple generations and structural analysis, how practical is the approach for real-world inference pipelines?

**Limitations:**

The paper acknowledges some limitations regarding the difficulty of accurately extracting reasoning structure and the limited scope of benchmarks used in the evaluation. Additional discussion of generalization to other reasoning domains and the computational cost of structural analysis would strengthen the limitations section.

**Strengths And Weaknesses:**

Strengths

•	Understanding how reasoning performance scales with inference-time compute is an important and active research topic. The paper directly engages with an ongoing debate regarding whether longer chain-of-thought reasoning improves performance. By analyzing multiple models and datasets, the work provides useful empirical evidence that challenges the simple “longer is better” narrative.

•	The proposed Failed-Step Fraction (FSF) metric is a novel attempt to quantify reasoning structure rather than surface-level properties of reasoning traces. By converting chain-of-thought traces into reasoning graphs and measuring the proportion of failed branches, the authors provide a potentially meaningful indicator of reasoning quality. The empirical results show that FSF correlates with accuracy more consistently than both length and review ratio across all ten models and both datasets.

•	The study includes ten reasoning models spanning multiple architectures and sizes (e.g., DeepSeek, Qwen, GPT-OSS, Claude). The analysis generates thousands of reasoning traces and uses conditional correlation techniques to control for question-level confounders. This broad empirical coverage strengthens the credibility of the observed patterns.

•	The authors attempt to test causality through two interventions (1) Test-time reranking, where reasoning traces are selected based on different metrics and (2) Controlled editing, where failed branches are removed from reasoning traces. These experiments provide interesting evidence that reasoning structure may influence performance rather than merely correlate with it.

Weaknesses

•	The reasoning graphs used to compute FSF are extracted using another LLM (Claude 3.7) that converts chain-of-thought traces into Graphviz representations. This introduces several potential concerns, namely (a) extraction accuracy depends on the behavior of the auxiliary LLM (b) failed-step labeling may reflect heuristic judgments rather than objective reasoning structure and (c) errors in graph extraction could propagate into the FSF metric. Although the paper claims high fidelity in graph generation, a more rigorous validation of the extracted graphs would strengthen the reliability of the metric.

•	The notion of a ‘failed branch’ is central to the paper’s claims, yet its definition remains somewhat ambiguous. A step is labeled as failed when it belongs to an abandoned reasoning branch, but determining whether a branch is abandoned may depend on subjective interpretation of the reasoning trace. The labeling process relies on model-generated annotations rather than ground-truth reasoning structures. As a result, FSF may partially reflect the judgment heuristics of the graph-extraction model rather than intrinsic properties of the original reasoning trace.

•	The experiments focus on HARP (math reasoning) and GPQA-Diamond (scientific reasoning). While both are challenging benchmarks, they represent relatively narrow reasoning domains. It remains unclear whether the observed relationships between FSF and correctness generalize to other types of reasoning tasks such as multi-hop question answering, program synthesis, or planning or interactive reasoning tasks.

•	Although the authors control for question-level effects using conditional correlations, residual confounding may still exist. For example, harder reasoning paths may naturally contain more exploratory branches, which would increase FSF and reduce accuracy simultaneously. In such cases, FSF could still function as a proxy for difficulty rather than a causal driver of reasoning quality.

•	The test-time reranking experiment demonstrates that selecting reasoning traces with lower FSF improves accuracy. However, this effect may arise simply because FSF acts as a proxy for partial correctness signals within reasoning traces. Similarly, editing reasoning traces to remove failed branches changes the input context and therefore modifies the model’s continuation behavior. These experiments provide suggestive evidence but do not fully establish causal mechanisms.

•	The proposed approach requires generating multiple reasoning traces, extracting reasoning graphs, and computing structural metrics for selection. This process may be computationally expensive for real-world applications where latency and cost are critical considerations.

---

> ### Author Rebuttal · Authors · 2026-03-31
>
> Thank you for the supportive evaluation and for recognizing the novelty of FSF as a structural metric, the breadth of our empirical study across ten models, and the value of our interventions in going beyond pure correlation. We address the main concerns below.
>
> **W1 (Graph extraction accuracy) and W2 (Failed branch definition ambiguity):** The reviewer notes that *"extraction accuracy depends on the behavior of the auxiliary LLM,"* that *"failed-step labeling may reflect heuristic judgments rather than objective reasoning structure,"* and that *"FSF may partially reflect the judgment heuristics of the graph-extraction model rather than intrinsic properties of the original reasoning trace."*
>
> Thank you for raising this. Due to the rebuttal word limit, we refer the reviewer to our response to Reviewer 3gMz (W1), where we address this point in detail.
>
> **W3 (Limited benchmarks):** The reviewer notes *"it remains unclear whether the observed relationships between FSF and correctness generalize to other types of reasoning tasks such as multi-hop question answering, program synthesis, or planning."*
>
> We have included additional benchmarks in the Appendix: MMLU Pro Computer Science and Engineering (80 questions, Figure 13, which covers coding adjacent and applied engineering reasoning). FSF shows significant correlations on MMLU Pro CS and Engineering as well. Combined with HARP (math, 300 questions) and GPQA Diamond (science, 198 questions), the evaluation spans math, science, CS, and engineering.
>
> **W4 (FSF as difficulty proxy):** The reviewer raises the concern that *"harder reasoning paths may naturally contain more exploratory branches, which would increase FSF and reduce accuracy simultaneously."*
>
> Our analysis is explicitly designed to address this. All correlations in our study are computed **within the same question, across different reasoning traces from the same model** (conditional correlation). Question difficulty is held constant by design. If FSF were merely a difficulty proxy, the within question correlation would vanish, but it remains the strongest and most consistent predictor across all models and benchmarks. This is the core purpose of the conditional design in Section 4.1.
>
> **W5 (Causal mechanism):** The reviewer notes that *"this effect may arise simply because FSF acts as a proxy for partial correctness signals within reasoning traces"* and that *"editing reasoning traces to remove failed branches changes the input context and therefore modifies the model's continuation behavior."*
>
> Thank you for this detailed critique. On FSF as a "proxy for partial correctness": our graph extraction pipeline contains **no correctness judgment**. The extraction prompt asks Claude 3.7 to label nodes as "successful" or "failed" *locally to the reasoning trajectory*, i.e., whether the model itself abandoned that branch, not whether the step is mathematically correct. A CoT can have every step labeled "successful" and still reach a wrong final answer. FSF measures structural exploration patterns, not partial correctness. Furthermore, Claude 3.7 Sonnet, the model used for graph extraction, is among the weakest models on the reasoning tasks we evaluate, yet it still produces FSF values with strong predictive and causal power across all 10 models (including much stronger ones like DeepSeek R1 and Qwen 3 235B). If FSF were merely capturing partial correctness signals, a weak extractor model would be poorly calibrated for stronger models' reasoning, but it is not.
>
> On "editing changes the input context": we respectfully clarify that this characterization is not accurate. In our editing experiment (Section 4.3), the *Reduced* condition keeps exactly the same prefix up to the point where the failed branch begins, then removes the failed branch and lets the model continue from that same prefix. The model's continuation starts from the same reasoning state.
>
> **W6 (Computational cost):** The reviewer notes that *"this process may be computationally expensive for real-world applications where latency and cost are critical considerations."*
>
> Please see our response to Reviewer fJTQ (W3 and on lower-cost variants).
>
> We thank the reviewer again for the careful reading and supportive assessment. We hope our responses have addressed your concerns, and we welcome any further questions.

---

> > ### Author Rebuttal · Reviewer_bFVd · 2026-04-04
> >
> > Thank you for your detailed rebuttal. My score is quite positive and I will keep that.

---

> > > ### Author Response · Authors · 2026-04-07
> > >
> > > Thank you for the positive follow-up and for maintaining your supportive score.

---

### Decision · Program_Chairs · 2026-04-30

**Decision:**

Accept (regular)

**Comment:**

This paper conducts a systematic evaluation across ten LRMs on math and scientific reasoning. It introduces a graph view of CoTs to extract structure and identify a single statistic—the Failed-Step Fraction (FSF), the fraction of steps in abandoned branches—that consistently out-predicts length and review ratio for correctness across models. It designs two interventions: it ranks candidate CoTs by each metric at test time, where FSF yields the largest pass@1 gains; second, it edits CoTs to remove failed branches, which significantly improves accuracy, indicating that failed branches bias subsequent reasoning.  The reviewers appreciated the simplicity yet the applicability of this work and its conclusions. The approach is clean and the experiments are well designed. However, the noisy graph extraction process  via a complex construction pipeline relies on prompting an LLM to convert CoT into Graphviz format, which is inherently a lossy and under-specified transformation.